# Defect-tolerant electron and defect-sensitive phonon transport in quasi-2D conjugated coordination polymers

Hio-Ieng Un [1] ✉, Kamil Iwanowski [2,3], Jordi Ferrer Orri [1,6], Ian E. Jacobs [1,6], Naoya Fukui [4], David Cornil [5], David Beljonne [5], Michele Simoncelli [2,3] ✉, Hiroshi Nishihara [4] & Henning Sirringhaus [1] ✉

Thermoelectric materials, enabling direct waste-heat to electricity conversion, need to be highly electrically conducting while simultaneously thermally insulating. This is fundamentally challenging since electrical and thermal conduction usually change in tandem. In quasi-two-dimensional conjugated coordination polymer films we discover an advantageous thermoelectric transport regime, in which charge transport is defect-tolerant but heat propagation is defect-sensitive; it imparts the ideal mix of antithetical properties—temperature-activated, exceptionally low lattice thermal conductivities of $0.2 \, W \, m^{-1} \, K^{-1}$ below Kittel's limit originating from small-amplitude, quasi-harmonic lattice dynamics with disorder-limited lifetimes and vibrational scattering length on the order of interatomic spacing, and high electrical conductivities up to $2000 \, S \, cm^{-1}$ with metallic temperature dependence, notably in poorly crystalline structures with paracrystallinity >10%. These materials offer attractive properties, such as ease of processing and defect tolerance, for applications, that require fast charge, but slow heat transport.

Structure determines properties and uses of materials. Electron and phonon motion as well as energy transfer processes are often faster and more efficient in crystalline materials than in amorphous solids. Single or poly-crystalline silicon have carrier mobilities three orders of magnitude higher than amorphous Si[1], and graphene has electrical (thermal) conductivity ten (three) orders of magnitude higher than glassy carbon[2,3]. To block the heat flow without significantly degrading the electronic transport, state-of-the-art inorganic and cage-like thermoelectric materials have been developed by introducing heavy oscillating atoms into the cages within crystalline structures to act as point scatterers and meanwhile reducing grain size below phonon but above carrier mean free paths. However, the lattice thermal conductivity of such crystalline materials often remains well above the

amorphous limit[4–8]. It is possible to reduce lattice thermal conductivities to very low values through porosity and anharmonicity approaches, such as porous metal-organic frameworks (MOFs) and organic semiconductors, respectively, but these materials are usually not highly electrically conducting[9–16]. In general, difficult trade-offs remain; finding materials with fast charge but slow heat transfer in a single structure is, therefore, fundamentally challenging but required.

In this work, we discover that the conjugated metal-organic framework (cMOF) or conjugated coordination polymer (cCPs), copper benzenehexathiol (Cu-BHT), is an unconventional material, in which electron motion in defective, non-crystalline films with para-crystallinity >10% can be delocalized and metallic, but lattice vibrations

[1]Optoelectronics Group, Cavendish Laboratory, University of Cambridge, Cambridge, UK. [2]Department of Applied Physics and Applied Mathematics, Columbia University, New York, NY, USA. [3]Theory of Condensed Matter Group, Cavendish Laboratory, University of Cambridge, Cambridge, UK. [4]Research Institute for Science and Technology, Tokyo University of Science, Noda-shi, Chiba, Japan. [5]Laboratory for Chemistry of Novel Materials, University of Mons, Mons, Belgium. [6]These authors contributed equally: Jordi Ferrer Orri, Ian E. Jacobs. ✉e-mail: hiu20@cam.ac.uk; michele.simoncelli@columbia.edu; hs220@cam.ac.uk

are rather localized. We carefully vary and quantify the degree of chemical and structural imperfection and relate it to the thermoelectric and magneto-transport coefficients. We show that Cu-BHT films exhibit an electrical conductivity/lattice thermal conductivity ratio $\sigma/\kappa_{latt}$ of up to $60 \times 10^4$ S K W$^{-1}$, which is 5–16 times higher than that of other state-of-the-art thermoelectric materials. This ideal mix of fast charge and slow heat transport is not observed in any other type of thermoelectric material, and implies an advantageous thermoelectric transport regime. Our work also provides guidelines to synthesis and addresses the question behind the unknown, huge electrical conductivity variation spanning over 0–2500 S cm$^{-1}$ that has been observed in Cu-BHT films in several studies using the same synthesis method (liquid-liquid interfacial synthesis)[17–20]. Our discovery of the ideal antithetical thermoelectric transport regime could overcome some of the technological challenges in thermoelectrics, since a high degree of crystallinity is no longer required, and thus provides a

potential pathway to enable higher performance, lower cost, and simpler processing than existing thermoelectric materials.

## Results

### Quantification of chemical and structural disorder

In the liquid-liquid interfacial synthesis of our Cu-BHT films we vary the molar ratio between the Cu precursor and the BHT in the growth solution from 2 to 7 around the nominally ideal ratio of 3 for perfect Cu$_3$BHT (Fig. 1a). Hereafter the ratio we added in the synthesis is referred to as the Cu/BHT ratio. In this way, we aim to influence the level of BHT and copper vacancies in the films and study the effects of the associated defects on the thermoelectric coefficients. Cu$_3$BHT has been considered to adopt a two-dimensional (2D), van der Waals layered structure[17–20], while, however, there is a very recent work[21] reporting a non-van der Waals layered structure. A comparison of the grazing-incidence wide-angle X-ray scattering (GIWAXS) patterns

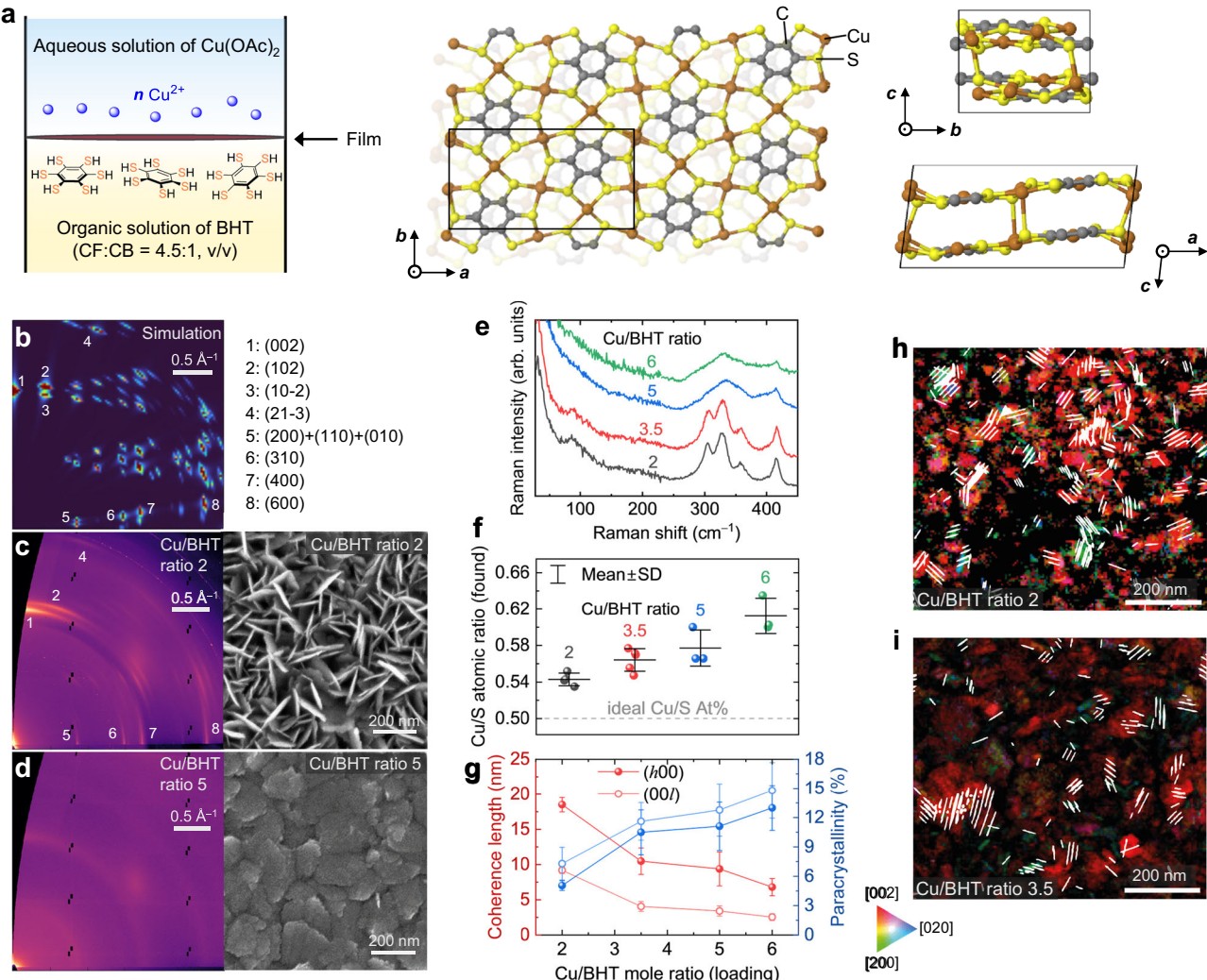

**Fig. 1 | Quantitative analysis of structural disorder and chemical defects in Cu-BHT films as a function of composition. a** Synthesis of Cu-BHT films by liquid-liquid interfacial method and diagram of the non-defective lattice of the Cu$_3$BHT structure reported recently[21]. *n* in the chemical reaction formula is the ratio of Cu$^{2+}$ to BHT added. **b** Simulated GIWAXS image of non-defective Cu$_3$BHT, with detailed peaks assignment in Supplementary Fig. 6. **c, d** Experimental GIWAXS and SEM images for a crystalline sample with Cu/BHT ratio of 2 and a more amorphous sample with a Cu/BHT ratio of 5, respectively. **e** Raman spectra of different compositions (see Fig. 3b for corresponding atom displacements). **f** Cu/S atomic ratios in the films detected by EDX as a function of Cu/BHT mole ratio (loading). The error bars represent the standard deviation of the results obtained from different local

positions of the films. **g** X-ray coherence length and paracrystalline disorder evaluated from WA analysis for the in-plane ($h$00) and out-of-plane (00 $l$) diffractions as a function of Cu/BHT ratio. The error bars reflect the uncertainties originating from the fitting of the diffraction peaks when extracting the peak positions and full widths of half maximum. **h, i** SED maps visualizing the stacking orientation of films with Cu/BHT ratio of 2 and ratio of 3.5 on the nanoscale. Face-on dominant grains (zone axis [002] mostly normal to the substrate) are highlighted in red, edge-on dominant grains (zone axes [200] and [020] mostly normal to the substrate) are highlighted in green and in blue, respectively. White streak lines correspond to grains with intermediate/mixed orientation. Black areas show the regions where diffraction is not good enough to estimate the zone axes.

(Fig. 1b, c and Supplementary Figs. 5 and 7) of our most crystalline films obtained from a Cu/BHT ratio of 2 with theoretical simulations indicates that our films are much more similar to and adopt the latter, newly reported non-van der Waals layered structure than the van der Waals layered structure. The distinct diffraction peaks that can be assigned as (102) and (21−3) and those located at $q_z = 0.6–1.2\,\text{Å}^{-1}$ and $q_r = 0.6–2.8\,\text{Å}^{-1}$, that are predicted for the newly reported structure, are clearly observed (Fig. 1b, c and Supplementary Fig. 6). Combined first-principles simulated and experimental GIWAXS with energy dispersive X-ray spectroscopy (EDX), scanning electron diffraction (SED), Raman spectroscopy, and X-ray Photoelectron Spectroscopy (XPS), the films we prepared are the non-van der Waals layered Cu3BHT with different ordering/defect levels (discussed later). The most crystalline films exhibit a pronounced face-on-preferred orientation, as evident by the (002) observed in $q_z$ direction, and films grown with higher Cu/BHT ratio are also mainly face-on, but exhibit larger regions with a more amorphous microstructure in between the face-on domains (Fig. 1b–d and Supplementary Fig. 10b–d). This is also apparent from the corresponding secondary electron microscopy (SEM) images. A gradually increasing coordination imperfection is also observed in Raman spectra (Fig. 1e): a broad, but well-resolved band centered at ca. 90 cm⁻¹, which is mainly attributed to Cu displacement, vanishes and a well-resolved triplet band at 305.0, 328.1, and 357.9 cm⁻¹, which is due to S displacement, loses its fingerprint and merges into a broad band as we move from low to high Cu/BHT ratio. SED in transmission mode was employed to directly visualize the stacking orientation on the nanoscale (Fig. 1h, i and Supplementary Figs. 11 and 13). A face-on stacking preference with the [002] zone axis normal to substrate (indicated in red), is seen to dominate in both Cu/BHT ratios of 2 and 3.5, while edge-on preference, i.e., zone axes [200] and [020] normal to substrate (green and blue, respectively), are less prominent. The black areas in Fig. 1h, i show scanned regions where the diffraction signal was not good enough to estimate the zone axes, presumably indicative of disordered or amorphous regions as well as grain boundaries; this result suggests that in addition to ordered grain boundaries, crystalline regions are possibly separated by more disordered/amorphous regions. EDX was performed to quantify the chemical composition and defect density (Supplementary Fig. 4). The actual Cu-to-S atomic ratio is positively correlated with the Cu/BHT molar ratio used during film growth (Fig. 1f). Under all growth conditions used the composition of the films remains Cu-rich, as evident by the Cu/S atomic ratio higher than the ideal value of 0.5, even for an excess of organic ligand BHT was added. These results suggest that the majority of chemical defects in our films are BHT vacancies and the densities are estimated to be one BHT vacancy in every 3, 2, 1.8, and 1.4 unit cells in the films prepared by Cu/BHT ratios of 2, 3.5, 5, and 6.5 during the synthesis, respectively.

To better understand the correlation between chemical defects and structural imperfections, as well as to quantify the contributions of different types of disorder − paracrystalline disorder $g$, strain-related lattice parameter fluctuations, X-ray coherence length (i.e., the crystallite size), pseudo-Voigt peak shape analysis[22] and fast-Fourier-transform (FFT) Warren-Averbach (WA) analysis[22,23] (Supplementary Fig. 10i−l) were applied. The analysis suggests long-range ordering in all compositions is paracrystallinity-dominated, with less important contributions from the strain-related lattice parameter fluctuations[22]. The X-ray coherence length, paracrystallinity, and strain-related lattice parameter fluctuations are calculated to be 18.5 ± 1.0 nm, 4.8 ± 1.2 %, and <1% for the intra-sheet lattice of the most crystalline, ordered Cu/BHT ratio of 2. As the Cu/BHT ratio increases, paracrystallinity and coherence length are found to gradually increase to 13% and reduce to <8 nm, respectively (Fig. 1g). Likewise, the parameter for the out-of-plane (00 $l$) ordering has similar compositional dependence. These results reveal that lower Cu/BHT ratio provides higher mean crystallite size and better long-range ordering. The transition from a crystalline to

a more disordered, quasi-amorphous structure is believed to be induced by chemical defects that distort the lattice.

## Defect-tolerant, metallic electron but defect-sensitive, glassy heat transport

The presence of these chemical and structural defects strongly influences the thermoelectric properties of the films. Unexpectedly, the most chemically perfect, crystalline composition (Cu/BHT ratio of 2) does not show the highest electrical conductivity, but only exhibits a value of 636 ± 245 S cm⁻¹ (Fig. 2c), while the more amorphous compositions (i.e., Cu/BHT ratio of 3.5–5.5) with paracrystallinity >10% and less prominent grain boundary features (Fig. 1c, g) exhibit electrical conductivities of up to ca. 2000 S cm⁻¹ (Fig. 2c). This is accompanied with a clear transition from a weakly thermally activated electrical conductivity (Cu/BHT ratio of 2) to a metallic behavior with conductivity increasing with decreasing temperature (Cu/BHT ratio of 5–5.5) (Fig. 2g, h). Correspondingly, the temperature-dependent Seebeck coefficient changes from a superlinear to a near linear temperature dependence behavior (Fig. 2f). Usually, metallic transport in many inorganic and organic (semi)conductors tends to occur in highly ordered crystalline or semicrystalline structures, while glassy systems with pronounced disorder (paracrystallinity >10%) often exhibit a strong temperature dependence of the electrical conductivity that drops by orders of magnitude from room to low temperature[1,2]. This is in stark contrast with our observation of a defect-driven metallic transition as well as the small drop in electrical conductivity from 300 K to ~25 K by only <20% even in the most disordered films (Fig. 2g, h). It is important to understand the origin of this disorder-tolerant charge conduction regime. Our attempts to fit the temperature dependent electrical conductivity with models for semi-localized transport[24], hopping between localized states[25,26] and trapping below extended states[27,28] were unsuccessful (Supplementary Figs. 26, 27, 29); these have been used to fit the temperature dependent electrical conductivity in other highly conducting materials, such as the coordination polymer NiTTFtt[29] or the organic conjugated polymers PEDOT[24], PBTTT[30] and PBFDO[31] in the (near) degenerate regime.

The observation of distinct grain boundaries by SEM and SED (Fig. 1c, h, i) suggests that electronic transport across grain boundaries is likely to be the limiting mechanism in the most crystalline films[32]. This mechanism is analyzed by using a heterogeneous transport model[33] that decomposes the measured electrical conductivity into contributions from intragrain metallic transport pathways and from disordered metallic and hopping transport pathways in the grain boundaries (Fig. 2h, i, Supplementary Fig. 28). The modeling shows clearly that the electrical conductivity of the crystalline films is indeed grain boundary limited, with the hopping energy between grains found to be 637 K (55 meV), and that the transport through the grain boundaries involves a disordered metallic pathway. It is possible that the local potential barriers associated with the grain boundaries could also induce an energy filtering effect (Fig. 2d), which selectively removes the contributions of low-energy charge carriers[32,34] and could potentially be the reason for the increase of the Seebeck coefficient in the most crystalline samples (Fig. 2b, f). In contrast, in the samples with higher paracrystallinity but less distinct grain boundaries, the energetic landscape becomes smoother. The associated, smoother energetic landscape contributes to the higher electrical conductivity.

Next, we analysed the lattice thermal conductivities of the different compositions (Fig. 2e), which can be calculated from the measured thermal conductivity by using the Wiedemann-Franz law which we find to be valid in this material (Supplementary Fig. 25). One of the most intriguing aspects of this material system is that significant structural disorder (paracrystallinity >10%) enables efficient, metallic charge motion but heat transport carried by lattice vibrations is pushed to the theoretical minimum limit (will be further discussed in the next section). The lattice thermal conductivity exhibits a mild

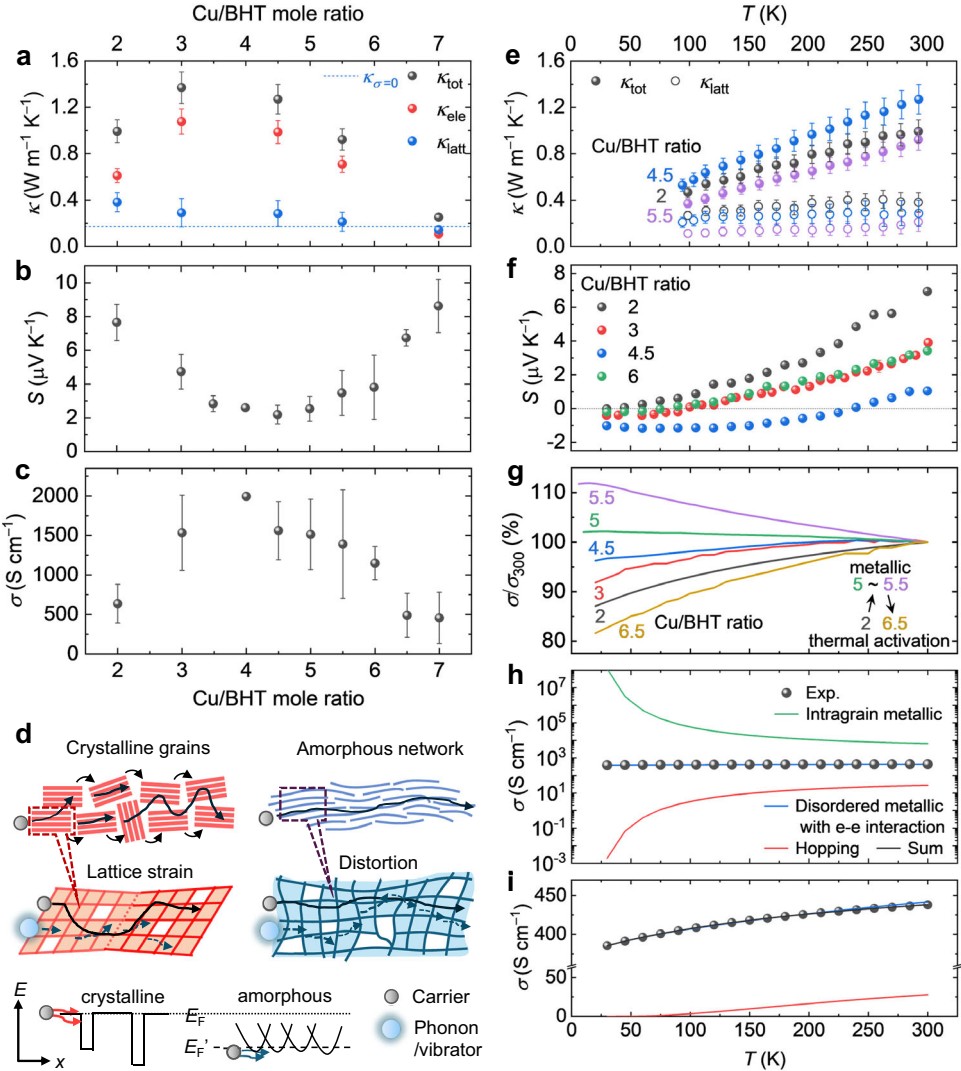

**Fig. 2 | Observation of defect-tolerant electron, but defect-sensitive heat transport in the temperature- and composition-dependent thermoelectric coefficients. a–c** Room temperature, in-plane thermal conductivities, Seebeck coefficients, and electrical conductivities of Cu-BHT films as a function of Cu/BHT ratio. In **a** our experimental thermal conductivity at the insulating state, i.e., lattice contribution at zero electrical conductivity, $\kappa_{\sigma=0}$ (blue dot line) is extracted from Supplementary Fig. 25. The error bars in Fig. 2b, c represent sample standard deviation. The data reported here are averaged over 90 devices for conductivities and 57 devices for Seebeck coefficient across 14 different batches of films. Among them, we only observed two devices showing negative Seebeck coefficient. **d** Diagrams illustrating the interplay between chemical and structural (im) perfections, coherent charge carrier and incoherent lattice vibrations, and electronic landscape. **e, f** Temperature-dependent in-plane thermal conductivities and Seebeck coefficients of different compositions. Since the Wiedemann-Franz law is

found to be valid (Supplementary Fig. 25), in **a, e** the electronic and lattice contributions to the total thermal conductivities are evaluated from the Wiedemann-Franz law. The error bars for total thermal conductivity in Fig. 2a, e reflect 7% uncertainty associated with the repeatability of the measurements, mainly due to the variability in thermal contacts and the measurement accuracy of the equipment. Combined with the ~7% uncertainty arising from measurement accuracy for the determination of electronic contribution, the error bars for the lattice thermal conductivity in Fig. 2a, e are obtained in quadrature. **g** Temperature-dependent electrical conductivity as a function of Cu/BHT ratio, where a transition from thermal activation to metallic transport is seen within a narrow range of compositions (ratio 5–5.5). **h, i** Decomposition of the measured electrical conductivity of Cu/BHT ratio of 2 into contributions from intragrain metallic transport pathway and from disordered metallic and hopping transport pathways in the grain boundaries.

increase with temperature, and even in the most crystalline composition, is only 0.38 W m$^{-1}$ K$^{-1}$. Such a record-low value observed in homogeneous, non-superlattice, non-porous crystalline solids with high atomic number density and extended networks of atoms bonded through non-weak interactions is indeed surprising; it is two orders of magnitude lower than the typical values of covalent solids[35] (>10 W m$^{-1}$ K$^{-1}$), comparable to the thermal conductivity of soft, van der Waals-bonded organic semiconductors[36–38] (<0.5 W m$^{-1}$ K$^{-1}$). In the more amorphous films with higher Cu/BHT ratio of 5.5, we observe even lower lattice thermal conductivity values down to 0.2 W m$^{-1}$ K$^{-1}$. The low thermal conductivity value is comparable to that of open, porous, crystalline MOFs with low atomic number density, such as non-

conjugated MOF-5 crystals[39]. Overall, our measurements in different Cu/BHT compositions highlight the coexistence of metallic, nearly free-electron-like electrical conductivity and record-low, thermally activated heat conductivity, exhibiting defect-tolerant electron, but defect-sensitive heat transport.

## Nature of vibrational heat carriers

In this section we focus on better understanding the lattice dynamics that underlies the defect-sensitive heat transport of Cu-BHT films. To this aim, we performed temperature-dependent Raman spectroscopy and GIWAXS measurements on the most crystalline composition (Cu/BHT ratio of 2). Temperature-dependent Raman

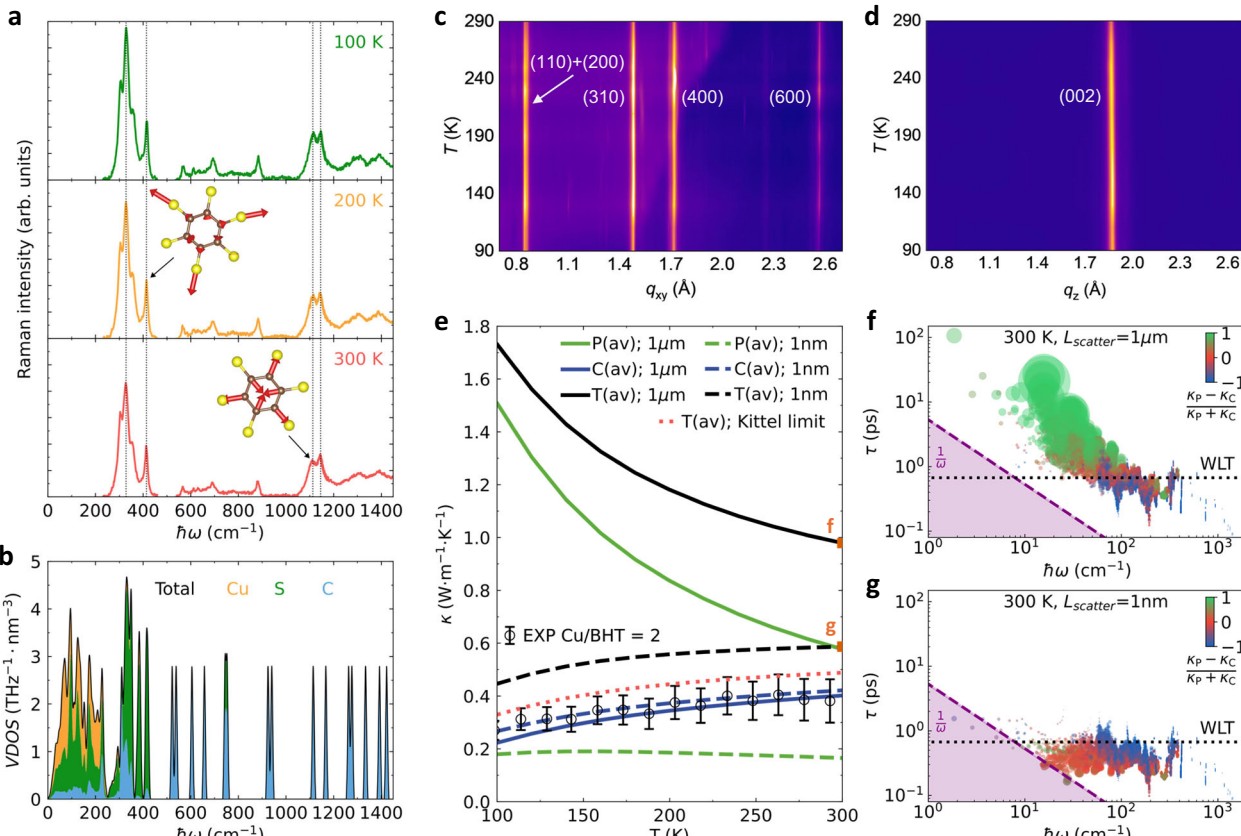

**Fig. 3 | Disorder-driven phonon-glass behavior in Cu-BHT films. a** Raman intensity as a function of temperature, with vertical lines showing negligible anharmonic shifts with temperature. The insets show the atomic displacement patterns of selected Raman-active modes. **b** The first-principles predictions for Vibrational Density of States (VDOS) resolved into contributions from Cu, S, and C atoms. **c**, **d** 2D maps of the temperature-dependent in-plane ($q_{xy}$) and out-of-plane ($q_z$) GIWAXS profiles, showing negligible peak shift and broadening in both directions. **e** Thermal conductivity predictions as a function of temperature for various scattering lengths: 1 μm, 1 nm, and the Kittel's theoretical limit (interatomic distance, -2 Å). "P" and "C" indicate conductivity contributions from particle-like

propagation and coherences' wave-like tunneling, respectively, and "T" is their sum (total conductivity). Empty black circles and the error bars are measurements of Cu/BHT ratio of 2 (same as in Fig. 2e, see there for the definition of the error bars). **f**, **g** Impact of scattering length on total vibrations' lifetime and contribution to conductivity−the circles' sizes are proportional to conductivity contributions, and their color shows whether they behave particle-like (green), wave-like (blue), or a mix of the two (red is 50% of each). "WLT" is the Wigner Limit in Time, a timescale determined by the material's structure that approximatively indicates at which lifetime vibrations transition from particle-like to wave-like behavior[43].

spectroscopy (Fig. 3a) provides evidence for only weak anharmonic effects in the lattice dynamics−from 4 to 300 K the Raman peaks only shift by <3 cm⁻¹, with Grüneisen parameter found to be 0.87, comparable to that of inorganic solids and other 2D materials. Temperature-dependent GIWAXS is able to unravel the anisotropic (in- vs cross-plane) lattice dynamics since thermal fluctuations can manifest themselves as changes in the intensity of Bragg peaks (non-cumulative disorder due to, for example, thermal vibrations) or peak broadening (cumulative disorder due to, for example, paracrystallinity)[22,40,41]. The uniaxial thermal expansion coefficients $\alpha$ are found to be $(10.7 \pm 0.80) \times 10^{-6} \, K^{-1}$ for the in-plane direction and $(38.3 \pm 0.83) \times 10^{-6} \, K^{-1}$ for the cross-plane direction (Supplementary Fig. 19), which is lower than organic semiconductors in most cases and more comparable to rigid inorganic and other 2D materials (Supplementary Table 2). Considering Cu-BHT's short inter-layer distance and its strong in- and out-of-plane chemical bonds, the force constants acting against thermal displacements in all dimensions are expected to be large. Consistently, both in-plane and cross-plane diffractions do not show apparent changes in intensity, line width, and line shape between 90 and 423 K (Fig. 3c, d and Supplementary Fig. 18), with pseudo-Voigt mixing parameter ($\eta$) remaining between 0.5 and 1 (Supplementary Fig. 18a, c) and only minimal shifts observed (Fig. 3c, d). These results suggest that in Cu-BHT, (1) large-

amplitude vibrations are suppressed and long-range ordering remains paracrystallinity-dominated (static disorder) upon thermal excitation and, (2) thermal fluctuations do not induce strong anharmonicity, pointing to a small-amplitude, quasi-harmonic lattice dynamics in all directions. This suggests that the observed low lattice thermal conductivity is not the result of strong anharmonicity effects.

To obtain a theoretical understanding of the exceptionally small lattice thermal conductivity, we performed first-principles simulations of heat conduction through lattice vibrations based on the Wigner Transport Equation (WTE)[42,43]. This framework generalizes the Boltzmann transport equation, accounting for the vibrational heat not only carried by phonons that propagate particle-like, but also by wave-like tunneling between vibrational eigenstates with energy difference smaller than their energy uncertainty (linewidth). As a result, the WTE offers a comprehensive approach to predict the lattice thermal conductivity of a wide range of materials, including ordered-and-anharmonic crystals[43], disordered-and-harmonic glasses[44], as well as the intermediate regime of disordered-and-anharmonic crystals[43].

For poor heat conductors such as Cu-BHT, it is accurate to consider the WTE solution in the relaxation-time approximation[43,45]. Moreover, to compare with experiments in polycrystalline disordered

samples where anisotropic transport could not be resolved, we consider the spatially averaged trace of the conductivity tensor,

$$\kappa(T) = \kappa_P(T) + \kappa_C(T) = \frac{1}{VN_c} \sum_{\mathbf{q},s} \left[ C(\mathbf{q})_s \frac{|\mathbf{v}(\mathbf{q})_{s,s}|^2}{3} [\Gamma(\mathbf{q})_s]^{-1} \right.$$
$$+ \sum_{s' \neq s} \frac{C(\mathbf{q})_s}{C(\mathbf{q})_s + C(\mathbf{q})_{s'}} \frac{\omega(\mathbf{q})_s + \omega(\mathbf{q})_{s'}}{2}$$
$$\left. \left( \frac{C(\mathbf{q})_s}{\omega(\mathbf{q})_s} + \frac{C(\mathbf{q})_{s'}}{\omega(\mathbf{q})_{s'}} \right) \frac{|\mathbf{v}(\mathbf{q})_{s,s'}|^2}{3} \frac{\frac{1}{2}[\Gamma(\mathbf{q})_s + \Gamma(\mathbf{q})_{s'}]}{[\omega(\mathbf{q})_{s'} - \omega(\mathbf{q})_s]^2 + \frac{1}{4}[\Gamma(\mathbf{q})_s + \Gamma(\mathbf{q})_{s'}]^2} \right] \quad (1)$$

Here, $C(\mathbf{q})_s = [\hbar^2 \omega^2(\mathbf{q})_s / k_b T^2] \bar{N}(\mathbf{q})_s [\bar{N}(\mathbf{q})_s + 1]$ is the specific heat at temperature $T$ of the vibration having wavevector $\mathbf{q}$, mode $s$, energy $\hbar\omega(\mathbf{q})_s$, and population given by Bose-Einstein distribution $\bar{N}(\mathbf{q})_s = \left[ \exp(\frac{\hbar\omega(\mathbf{q})_s}{k_B T}) - 1 \right]^{-1}$; $\Gamma(\mathbf{q})_s$ is the total linewidth that accounts for anharmonic phonon-phonon scattering, electron-phonon scattering, presence of isotopic impurities, and scattering with grain boundaries (see Supplementary Section 4.1 for details). $\mathbf{v}(\mathbf{q})_{s,s'}$ is a velocity matrix —its diagonal elements $s = s'$ are the phonon group velocities, and its off-diagonal elements describe the strength of coherence's couplings between modes $s$ and $s'$ at the same $\mathbf{q}$; $N_c$ is the number of wavevectors sampling the Brillouin zone and $V$ is the crystal's unit-cell volume. The total conductivity (1) accounts for both contributions from particle-like propagation ($\kappa_P$) and coherence' wave-like tunneling ($\kappa_C$) to the heat transport, respectively. These macroscopic conductivities can be resolved in terms of microscopic, single-vibration contributions, as shown by the two terms inside the square brackets: the first one describes phonons that carry heat $C(\mathbf{q})_s$ by propagating particle-like with velocity $\mathbf{v}(\mathbf{q})_{s,s}$ over lifetime $[\Gamma(\mathbf{q})_s]^{-1}$; the second one, instead, accounts for wave-like tunneling involving pairs of phonons $s, s'$ at the same wavevector $\mathbf{q}$. It has been shown that in ordered and weakly anharmonic crystals particle-like propagation dominates over wave-like tunneling ($\kappa_P \gg \kappa_C$)[42], in complex crystals both these mechanisms co-exist ($\kappa_P \sim \kappa_C$), and in strongly disordered glasses around room temperature, tunneling dominates ($\kappa_P \ll \kappa_C$)[46].

Our predictions for the thermal conductivity (Fig. 3e) suggest that nearly pristine, crystalline Cu-BHT (with micrometric or larger grains, i.e., scattering length of 1 μm) should feature a total conductivity that decreases with increasing temperature (crystal-like trend) between 100 and 300 K. This originates from having, in these ideal, ordered samples, particle-like transport mechanisms stronger than wave-like ones (Fig. 3f). We also note that at 300 K the conductivity of nearly pristine Cu-BHT is predicted to be 1 W m⁻¹ K⁻¹ (Fig. 3e), a value that is still orders of magnitude lower than the conductivity of typical dense inorganic materials, highlighting the poor intrinsic lattice heat conduction in Cu-BHT. Such a property originates from the heavy copper atoms, which contribute to phonon modes with energy below 250 cm⁻¹ (Fig. 3b) and propagate heat poorly; this can be understood by noting that these modes feature low group velocity and relatively short (still non-overdamped[43,45]) lifetime, conditions that imply a small contribution to the lattice thermal conductivity (see Eq. 1). Accounting for structural disorder, with phonon scattering at the nanometric scale (using the Casimir model[47], see Supplementary for details) yields dramatic changes in the macroscopic conductivity: (1) a strong reduction of the overall magnitude; (2) an inversion of the trend in temperature, from decreasing with $T$ (crystal-like) to increasing with $T$ (glass-like). Figure 3g demonstrates that these macroscopic changes result from the different effects of phonon-disorder scattering on microscopic heat transport mechanisms; in particular, while disorder suppresses particle-like propagation, it has a negligible impact on wave-like tunneling, making the latter the dominant conduction mechanism in the presence of strong disorder. Finally, we highlight how

considering the theoretical minimum for the phonon scattering length due to disorder—the average bond length (~2 Å), also known as Kittel's limit[48]—yields predictions compatible with experiments (dotted-red line in Fig. 3e).

Our combined experimental and theoretical analyses therefore reveal that Cu-BHT films at and below room temperature feature: (1) quasi-harmonic lattice dynamics; and (2) disorder-driven, ultra-low temperature-activated lattice thermal conductivity. Specifically, structural disorder—likely a mixture of polycrystalline, paracrystalline, or amorphous domains—dominates over phonon-phonon and electron-phonon interactions in determining the conductivity. This is evidenced microscopically by the predominance of wave-like tunneling over particle-like propagation conduction mechanisms, leading to the thermal conductivity being strongly limited by the structural defects.

## Nature of charge carriers

Low energetic disorder in strongly electronically coupled materials with weak electron-phonon coupling promotes wave function delocalization and electron-crystal transport behavior, which can be probed through magnetotransport measurements. The magnetoresistance (MR), defined as $[\rho_{xx}(\mathbf{B}) - \rho_{xx}(0)]/\rho_{xx}(0)$, was found to be positive, exhibit a quadratic-field-dependence at all temperatures from 10 to 300 K in both crystalline Cu-BHT and films with paracrystallinity >10% (Fig. 4b, c) and was much larger for a transverse orientation of the magnetic field with respect to the current direction than for a longitudinal orientation (Fig. 4a). The MR can be understood by carrier motion in a metallic regime; it can arise even in a unipolar transport regime due to anisotropies in the band structure, but is most commonly interpreted as a signature of an ambipolar transport regime, in which both electron and hole pockets on the Fermi surface contribute to transport, but exhibit different Hall angles[49]. The most crystalline sample (Cu/BHT ratio of 2) exhibits a lower transverse MR and weaker angular dependence at 14 T than the disordered, near-amorphous sample (Cu/BHT ratio of 5; Fig. 4a–c and Supplementary Fig. 28). The difference is smaller than what would be expected from Kohler's rule, which states that the magnitude of the MR is determined by the magnitude of the magnetic field scaled by the zero field resistance $B/R$ (0, $T$). Kohler's rule is also found to be violated when analyzing the temperature dependence of the MR (insets in Fig. 4b, c). This suggests that the observed differences in the transport properties of crystalline and amorphous samples, as well as the temperature dependence of the MR do not merely reflect variations in carrier relaxation times, but more substantial changes in the transport physics.

To further elucidate this, Hall effect measurements were performed. Interestingly, the temperature-dependent Hall coefficient $R_H$ (Fig. 4d and Supplementary Fig. 33) is positive, i.e., hole-dominated, at all temperatures and exhibits a peak at around 135 K for the metallic composition with paracrystallinity >10%. In contrast, in the most crystalline films, we observed a sign change of the Hall coefficient around 210 K, with transport at higher temperatures becoming electron-dominated. We note that the Hall results here might not be directly quantitatively comparable to the results of Seebeck coefficient, which shows a hole-dominated behavior near room temperature. The Seebeck and Hall/MR measurements were not performed on the same samples, and the device preparation and loading of the devices for magnetotransport measurements required some unavoidable aging upon light and air exposure, which could have caused a shift of the Fermi energy within the electronic structure. However, it is still clear that there is a sign change observed in the Seebeck coefficient as a function of temperature, as well, most prominently in Cu/BHT ratio of 4.5. These observations therefore strongly suggest that a two-band transport model with contributions from both electron and hole pockets within a metallic electronic structure[50,51] does indeed

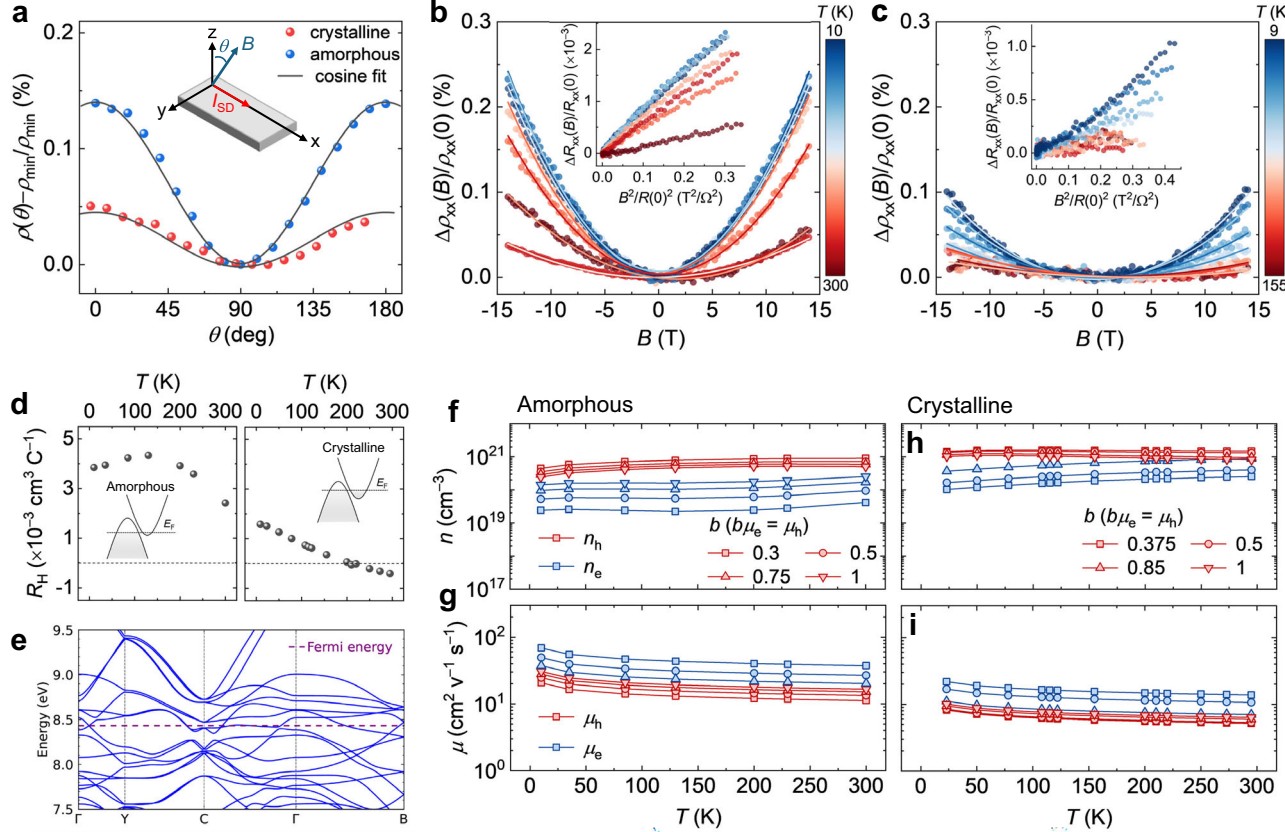

**Fig. 4 | Defect-manipulated anisotropic, electron-crystal magnetotransport in crystalline and near amorphous Cu-BHT films. a** Angular-dependent magnetoresistance measurements at 14 T and 50 K showing minima at $\theta \sim 90$ for both amorphous Cu/BHT ratio 5 and crystalline ratio 2. The architecture of the measurements is illustrated in the inset. **b, c** Quadratic magnetoresistance for amorphous and crystalline samples respectively ($\theta = 0$). Corresponding Kohler's plots are shown in the inset. **d** Temperature-dependent Hall coefficient of amorphous (left) and crystalline (right) compositions, with inset diagrams illustrating different Fermi levels at two-band transport region. **e,** Calculated electronic band structure for the newly reported non-van der Waals layered structure shown in Fig. 1a. **f, g** Simulated temperature-dependent carrier concentrations for holes and electrons and their mobilities of the amorphous sample. **h, i** Extracted temperature-dependent carrier concentrations for holes and electrons and their mobilities of the crystalline sample. $b$ is a scaling factor, $b\mu_e = \mu_h$.

need to be considered to consistently explain the MR and Hall data (Fig. 4e).

In the disordered composition, the UPS measurements indicate a comparatively high work function (Supplementary Fig. 16), possibly due to p-type doping by the defects that are present. The Fermi level is presumably at a position within a hole pocket but near to the bottom of an electron pocket, so that below the peak in Hall coefficient at 135 K transport is hole dominated, but above a thermally generated population of electrons is beginning to occupy the electronic states of the electron pocket and contribute to the transport. In the crystalline samples, the work function is reduced, i.e., the Fermi level shifts up to a position where it crosses both bands. In this regime, a hole-electron competing transport is at play. This is supported by our simulated electronic band structure of Cu-BHT, which shows both electron and hole pockets contributing at the Fermi level (Fig. 4e).

Based on these results, we developed a simple two-carrier transport model that is consistent with the temperature-dependent conductivity, MR, and Hall data, and allows us to estimate hole and electron carrier concentrations $n_h$, $n_e$, and mobilities $\mu_h$, $\mu_e$. In principle, this is an underdetermined problem, and we need to make certain approximations in the model. We assume, in particular, that the ratio of hole and electron mobility $b$ is a temperature-independent constant (see Supplementary Section 6). The simulated results show some clear and consistent trends, that is, $\mu_e > \mu_h$ and $n_e < n_h$, for both amorphous and crystalline samples:

(i). Sample with paracrystallinity >10%: $\mu_e = 17 - 37 \text{ cm}^2\text{V}^{-1}\text{s}^{-1}$; $\mu_h = 11 - 17 \text{ cm}^2\text{V}^{-1}\text{s}^{-1}$; $n_e = 0.42 - 2.6 \times 10^{20} \text{ cm}^{-3}$; $n_h = 5.1 - 9.1 \times 10^{20} \text{ cm}^{-3}$;

(ii). Crystalline composition: $\mu_e = 6.4 - 13.8 \text{ cm}^2\text{V}^{-1}\text{s}^{-1}$; $\mu_h = 5.2 - 6.4 \text{ cm}^2\text{V}^{-1}\text{s}^{-1}$; $n_e = 0.26 - 1.07 \times 10^{21}$; $n_h = 0.82 - 1.5 \times 10^{21} \text{ cm}^{-3}$

The ranges indicated reflect uncertainties in the approximations made in the model. For comparison, $1.05 \times 10^{21} \text{ cm}^{-3}$ corresponds to 2 carriers per primitive cell. We attribute the higher mobilities of disordered, near amorphous samples to a smoother energetic landscape that less significantly scatters holes and traps electrons. The observation of weakly increasing $\mu_e$ and $\mu_h$ with decreasing temperature in both crystalline and near amorphous samples consistently suggests that carrier motions in Cu-BHT are mainly in a (disordered) metallic regime, that is, defect-tolerant.

## Discussion

We have investigated the structure-property relationships and thermoelectric transport physics of Cu-BHT, as a model system for 2D, conjugated coordination polymers. We have found the charge transport to be defect-tolerant—metallic electrical conductivities > 2000 S cm$^{-1}$ can be reached in films that contain a high density of structural defects and exhibit paracrystallinity values >10%. This defect tolerance sets it apart from polymer (semi)conductors in which electron transport is defect-suppressed (Fig. 5a) and in which the electrical conductivity typically exhibits a non-metallic temperature dependence

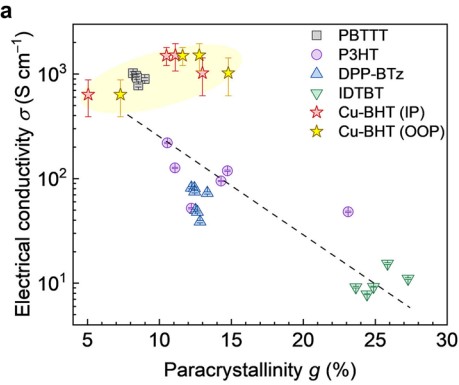

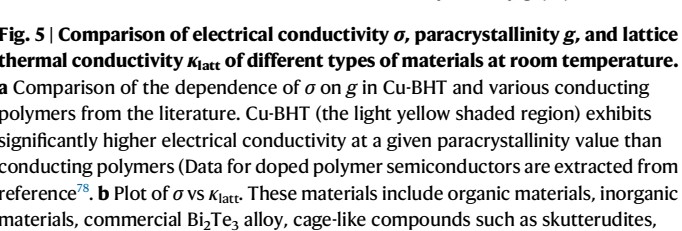

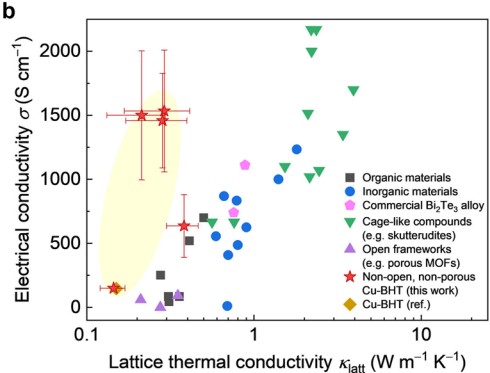

**Fig. 5 | Comparison of electrical conductivity $\sigma$, paracrystallinity $g$, and lattice thermal conductivity $\kappa_{latt}$ of different types of materials at room temperature.** **a** Comparison of the dependence of $\sigma$ on $g$ in Cu-BHT and various conducting polymers from the literature. Cu-BHT (the light yellow shaded region) exhibits significantly higher electrical conductivity at a given paracrystallinity value than conducting polymers (Data for doped polymer semiconductors are extracted from reference[78]. **b** Plot of $\sigma$ vs $\kappa_{latt}$. These materials include organic materials, inorganic materials, commercial $Bi_2Te_3$ alloy, cage-like compounds such as skutterudites, open frameworks such as porous MOFs, and non-open, non-porous Cu-BHT (this

work and previous work). The values for our Cu-BHT, indicated by stars in the light yellow shaded region, highlight the advantageous defect-tolerant charge and defect-sensitive phonon transport regime discovered in this work that provides an electrical conductivity/lattice thermal conductivity ratio $\sigma/\kappa_{latt}$ of up to $60 \times 10^4$ S K $W^{-1}$ that is 5–16 times higher than that of other state-of-the-art thermoelectric materials. The figure does not include materials in which thermoelectric enhancement is attributed to using special techniques such as nanostructuring or alignment. Detailed data from the relevant references[4–16,19,79,80] are provided in Supplementary Table 3.

$(d\sigma/dT > 0)$ at low temperatures. Importantly, this defect-tolerant electron transport in Cu-BHT coexists with defect-sensitive heat transport with a lattice thermal conductivity that is exceptionally low and practically at the theoretical limit. We have shown that this originates from having low-amplitude lattice vibrations, which are quasi-harmonic, but for which the associated thermal conductivity is predominantly wave-like. We have shown that the thermal conductivity of Cu-BHT is expected to be intrinsically low, even in perfect single crystals, and is further suppressed strongly by defects and damped by structural disorder, in sharp contrast with the defect-tolerant behavior observed for electrons. This ideal mix of antithetical properties in disordered Cu-BHT is analogous to, but distinct from, the electron-crystal, phonon-glass behavior that is observed in some highly crystalline inorganic thermoelectric materials. This advantageous thermoelectric transport regime in non-crystalline structures, providing an electrical conductivity/lattice thermal conductivity ratio $\sigma/\kappa_{latt}$ of up to $60 \times 10^4$ S K $W^{-1}$ that is 5–16 times higher than that of other state-of-the-art thermoelectric materials (Fig. 5b), remains largely unexplored, but could be practically significant since we no longer need structural perfection to achieve high performance. On the basis of magneto-transport measurements as well as first-principles calculations, we have argued that Cu-BHT exhibits a hole-electron competing ambipolar transport regime that prevents the Seebeck coefficient from reaching competitive levels. At present, the thermoelectric performance of our Cu-BHT is credible, but not particularly high with $ZT = 0.0015$ and power factor $\sigma S^2 = 5.0$ µW m$^{-1}$ K$^{-2}$; this is comparable to that of many conducting polymers. However, our findings suggest that techniques to tune the Fermi level into an effective, one-band regime and control the unipolar carrier concentration could be effective for enhancing the Seebeck coefficient and thus thermoelectric performance. This may be regarded as analogous to the development of polymer thermoelectrics, where enhancing thermoelectric performance also required advanced (de)doping methods to boost the Seebeck coefficient[10]. In any case, conjugated coordination nanosheets could offer attractive properties, such as ease of processing and defect tolerance, for applications, that require fast charge, but slow heat transport, which are not limited to thermoelectrics, but also include multi-functional thermal barrier coatings, as required in thermionic devices, some wearable electronic applications, integration of conducting sensors into building insulation or aerospace and energy applications[52–56].

## Methods

### Synthesis

Adapting literature recipes[18,20], we improve the liquid-liquid interfacial synthesis of Copper benzenehexathiol (Cu-BHT) in this work. The organic solvent used here is a mixture of chloroform (CF) and chlorobenzene (CB) in a ratio of CF:CB = 4.5:1, v/v. Under nitrogen atmosphere, BHT is first dissolved in degassed organic solvent at room temperature to provide a solution with a concentration of 1.11 mM. After filtering through a 0.45 µm PTFE membrane, BHT solution of 1 mL is added into a small vial (20 mL volume) containing mixed organic solvent of 4.5 mL. Filtered, degassed water of 6.5 mL is then added onto the organic phase to form an interface. Depending on the desired Cu/BHT ratio, certain amount of aqueous solution of Cu(CH$_3$COO)$_2$ in a concentration of 5 mM is filtered and added onto the water at 45 °C very gently (not to fluctuate the oil-water interface). Lastly, the reaction system is sealed and left standing at 45 °C for 2 h in nitrogen.

To transfer the Cu-BHT films onto substrates for characterizations, first, the upper layer, aqueous solution, is partly removed and diluted, washed with water very gently without perturbing the interface, in order to remove residual Cu$^{2+}$ and water-soluble impurities. Then, an O$_2$ plasma-cleaned, facing down substrate passes through the interface; at this point, the Cu-BHT film attaches on it. The substrate with the film is held on in the organic phase, simultaneously the aqueous layer is fully removed to expose the surface of the organic solution. Finally, the substrate with a film on it is slowly taken out. After naturally drying out, the substrate with film attached can be immersed into organic solvents, ethanol, followed by CF, for washing without lift off (except for methanol, which leads to lift off). These specific procedures usually generate films with thickness ranging from 100 to 400 nm, depending on the amount of Cu$^{2+}$ added.

The synthesis improved in this work enables anisotropic Cu-BHT films, different from previous study reporting isotropic films[20]. Note that the purity of the starting material BHT is significant for the electronic properties of the Cu-BHT films. Light yellow BHT that contains impurities impedes electronic conduction to enter a metallic regime (see Supplementary Section 5.5 for details). All samples investigated in this work follow the above procedures unless otherwise specified.

## Scanning electron microscopy (SEM) and energy dispersive X-ray spectroscopy (EDX)

The surface morphologies of the Cu-BHT films were imaged using a Zeiss LEO 1550 field-emission SEM with a working distance of 3.5–4 mm, an acceleration voltage of 3 kV, and an in-lens detector. The element composition was acquired using a dual-beam microscope Helios FIB-SEM equipped with Oxford Instruments EDS detector at the cross-section of the films. An acceleration voltage of 20 kV, working distance of 4 mm, and an acquisition time of 3–5 min were used during EDS measurement, where no drift was seen. Samples for these two measurements were deposited on Si without thermal-growth oxide.

## Grazing-incidence wide-angle X-ray scattering (GIWAXS)

GIWAXS measurements were performed at Diamond Light Source beamline I-06 at 12.5 and 20 keV X-ray beam energy for room and low temperature measurements, respectively. Samples were prepared on $1 \times 1$ cm Si wafer without thermal-growth oxide. Images were collected using a Pilatus 2 m camera positioned 450.5 mm from the sample. Sample-detector distance was calibrated using a silver behenate reference sample. All measurements were performed at an incidence angle of 0.2°, with samples mounted on a temperature-controlled stage inside a helium-filled chamber with Kapton windows for room temperature measurement and under vacuum for low temperature measurement. Temperature-dependent measurements consisted of two 5 s exposures at attenuation level 2 (9.9% of full beam intensity) with shifted camera position to gap-filled regions between detectors. A full sample realignment was performed between temperature steps to ensure consistent sample-detector distance, which could otherwise lead to systematic errors in the measured reciprocal spacing $q$. Data was processed using the MATLAB package GIXSGUI[57].

## Scanning electron diffraction (SED)

To fabricate thin, electron-transparent Cu-BHT films, the reaction time was shortened to 15–20 min, yielding films of 40–70 nm. The resulting thin films were transferred onto $O_2$-plasma-cleaned $SiN_x$ grids with a 30 nm-thick, low-stress amorphous $Si_3N_4$ membrane window (NT025X, Norcada). Care was taken during transferring, and optical microscope and SEM were employed to check the samples. The procedures of acquiring SED microscopy and the data processing are reported in previous work[58]. During acquisition, a two-dimensional (2D) electron diffraction pattern was measured at every probe position of a scanning electron beam in transmission mode. SED data were acquired on the JEOL ARM300CF E02 instrument at ePSIC (Diamond Light Source, Didcot-Oxford, UK). An acceleration voltage of 200 keV, nanobeam alignment (convergence semiangle) of 1 mrad, electron probe of 5 nm, a scan dwell time of 1 ms, a fluence of ~11 $e^- Å^{-2}$, and camera length of 20 cm were used. Post-processing of SED data was done using pyXem 0.14[59] (an open-source Python library for crystallographic electron microscopy). All diffraction patterns were distortion-corrected and calibrated with an Au cross grating. The drift in the beam position of the non-scattered beam was corrected and centered for all frames using cross-correlation with a subpixel factor of 10. To display diffraction planes that match real-space features, all diffraction patterns shown in this work were rotation-corrected using a $MoO_3$ calibration sample. Dead pixels and detector junctions were masked. The analysis of the processed SED data was done using the Automated Crystal Orientation Mapping in py4DSTEM 0.13.6[60]. To obtain the orientation map, peak finding through template matching was run on each probe position and was compared to a simulated library (from the CIF file) using sparse correlation matching. The flow map was generated by radially binning diffraction patterns in the range of 0.1 to 1.25 $Å^{-1}$ and obtaining the strongest angular direction. Virtual dark field images were generated from selecting specific diffraction spits and mapping its intensity as a function of probe position.

## Raman spectroscopy

Raman spectra were collected by using a Horiba T64000 Raman spectrometer under a laser excitation of 532 nm. Samples were prepared on either $1 \times 1$ cm Si wafer without thermal-growth oxide or Corning EAGLE XG glass. Before acquisition, the spectrometer was calibrated with the characterized band (520.70 $cm^{-1}$) of a standard Si sample. Regular measurements were performed in ambient air at room temperature, and temperature-dependent measurements were under high vacuum with liquid Helium as cooling system. During acquisition, integration time of no <30 s, 10 cycles, and a laser power that did not lead to visible beam damage were used.

## Ultraviolet photoelectron spectroscopy (UPS)

UPS experiments for measuring valence bands and work functions were carried out on the films transferred onto n-doped Si without thermal-growth oxide by using an ultrahigh-vacuum photoemission instrument, Escalab 250Xi, with a 21.22 eV excitation source. The samples had not been exposed to air by using a transfer tube for sample transferring and loading. During scanning an electrical bias of −5 V was applied to the samples. Scanning steps of 0.05 and 0.01 eV were respectively used for full-range spectra showing the secondary electron cutoff and, for the narrow, near-zero binding energy range, which shows the edge of the valence bands.

## Device fabrication

For initial electrical conductivity and Seebeck coefficient measurements, a four-parallel-electrode device architecture was used (Supplementary Fig. 1a). For all temperature-dependent measurements—electrical conductivity, Seebeck coefficient, DC and AC Hall effects, and magnetotransport—a multifunctional, integrated device architecture[61,62] was employed (Supplementary Fig. 1b). Substrates (Corning EAGLE XG, thickness ~700 µm) were cleaned by sequential sonication steps in acetone, 2% Decon 90 / DI water, DI water, and isopropanol (10 minutes for each). Then washed substrates were dried with nitrogen gas and exposed to oxygen plasma at 300 W for 3–5 min. All electrical contacts, Cr/Au (4 nm/20 nm), were deposited on the freshly prepared, cleaned substrates by thermal evaporation through shadow mask method for the four-parallel-electrode devices and patterned by UV lithography for multifunctional devices. Cu-BHT films were transferred onto the substrates with patterned electrodes as described in the synthesis.

## Room-temperature electrical conductivity and Seebeck coefficient measurements

Measurements were performed on a manual probe station using an Agilent 4155B Semiconductor Parameter Analyzer for electrical conductivity measurement and using a Keithley Nanovoltmeter 2182 A for Seebeck coefficient measurement under nitrogen atmosphere (Belle Technology, <5 ppm $O_2$ and <15 ppm $H_2O$). Each device was electrically isolated before measurement by carefully scratching off the film outside the active device area under an optical microscope. Film thickness was measured by surface profilometry.

## Temperature-dependent electrical conductivity and Seebeck coefficient measurements

For temperature-dependent thermoelectric measurements, data were recorded on multifunctional devices in a LakeShore Cryotronics CRX-4K probe station equipped with closed-cycle liquid Helium under high vacuum ($10^{-7}$ to $10^{-6}$ mbar). A Keithley 2182 A nanovoltmeter (for thermovoltage determination) and a couple of Keithley 2612B sourcemeters (for voltage sourcing and four-probe conductivity measurements) were used. The methodology of the Seebeck coefficient measurement is adapted from previous work by Venkateshvaran et al.[61] and by Statz et al.[62]

## Room-temperature and temperature-dependent thermal conductivity measurements

Cu-BHT films were deposited onto specially designed, commercially available silicon-based chips (referred to as Linseis chips), which are ordered from Linseis Messgeraete GmbH Vielitzerstr (Distributor: Gammadata UK Limited). The device architecture can be found in Supplementary Fig. 2. The chips containing two free-standing, different-area $Si_3N_4$ membranes. Two microfabricated electrical wires aligned with the longitudinal axes of the membranes serve as heater and thermometer with ALD-growth $Al_2O_3$ covering on the top as passivation layer. A high width/thickness ratio of the membranes on which Cu-BHT films covered ensures that the heat flux is predominantly horizontally one-dimensional across the sample and in line with the plane of the membranes. Thus, the measurements performed in this work probe the in-plane thermal conductivities of the samples by using a $3\omega$-based method with the Linseis Thin Film Analyzer (TFA). By performing measurements on the two different-area membranes integrated on the same substrate, a correction for radiative losses is made, and the contribution from empty membranes to total heat conduction (from membrane and sample) is also taken into account and subtracted; accurate measurements are hence enabled. Calculation of thermal conductivity was carried out on the raw data in the software of Linseis TFA according to the method developed by Linseis et al. [63] In addition to this reference by Linseis et al., detailed information about the measurement capacities of the instrument Linseis TFA, the structure of the Linseis chips, the working principle of the thermal conductivity measurement, and the measuring range and instrumental accuracy can be found here: https://www.linseis.com/en/instruments/thin-film-thin-film-analysis/tfa/.

## Temperature-dependent DC Hall effect and magnetoresistance measurements

DC Hall effect and magnetoresistance measurements were carried out on Quantum Design 14T-PPMS DynaCool D-134 with either standard or rotator sample pucks. Multifunctional Hall bar device architecture was used, and electrical connection between the device and the sample puck was built by wire bonding. Four-point-probe electrical conductivity at zero field was also recorded as a function of temperature during the measurement.

## First-principles predictions for the vibrational and thermal properties and the electronic band structure

Since past works discussed the layered Cu-BHT structure to naturally form in the AA stacking pattern, our simulations started comparing the energy of the different known structures for AA stacked layered Cu-BHT: (1) the established AA stacked structure, which contains 15 atoms per primitive cell[17,18,20], and (2) the recently observed non-van der Waals layered structure that contains 60 atoms per primitive cell[21]. Starting from the experimental primitive cells[21], we relaxed atomic positions and lattice vectors using Density Functional Theory (DFT) as implemented in Quantum Espresso[64,65]. We employed the PBEsol functional with Grimme-D2 van der Waals corrections, kinetic energy cutoff for charge density and wavefunctions equal to 720 Ry and 90 Ry, respectively, $6.6(6) \times 10^{-7}$ Ry/atom energy threshold, and $6.6(6) \times 10^{-7}$ Ry/Bohr/atom force threshold, and 0.05 kbar pressure convergence threshold. A Marzari-Vanderbilt smearing equal to 0.02 Rydberg was used. The Brillouin zones were sampled using a Monkhorst-Pack meshes equal to $4 \times 4 \times 8$ and $2 \times 4 \times 4$ for 15 and 60 atom structures, respectively, with a zero shift for both structures. We employed pseudopotentials taken from the SSSP efficiency library[66].

After complete relaxation, the energy per atom of the 15-atom cell ($-97.5625$ Ry/atom) resulted higher than that of the 60-atom cell ($-97.5634$ Ry/atom), indicating that the latter is energetically favored at this level of theory. We have also verified that performing calculations at the PBEsol + Grimme-D3 level yields consistent results ($-97.5590$ Ry/atom and $-97.5599$ Ry/atom for 15 and 60-atom structures, respectively). Given that the 60-atom cell was found to be energetically favored[67], we focused on it in the remainder of the analysis. In addition, given the unimportant differences between the PBEsol+D2 and PBEsol+D3 level of theory, and accounting for the fact that in Quantum Espresso Density Functional Perturbation Theory (DFPT, needed to compute the electron-phonon contribution to the linewidth $\Gamma_s(q)$ appearing in Eq. 1) is implemented at the PBEsol+D2 but not PBEsol+D3 level, we proceeded using DFPT with PBEsol+D2.

Second and third-order force constants were obtained using hiphive[68] in the 60-atom cell with 120 rattled structures; we used cutoffs equal to the default value 3.0 Å for both second- and third-order force constants. Electron-phonon couplings were calculated using DFPT, and given the large size of the primitive cell of non-van der Waals layered Cu-BHT (60 atoms), a Gamma-point ($q = 0$ only) calculation was performed. The Wigner conductivity expression (1) was evaluated accounting for electron-phonon[69,70], phonon-phonon[71,72], phonon-isotope[73], and phonon disorder scattering[47] (see Supplementary Section 4.1 for details). Electron-phonon linewidths on a dense mesh were obtained using Wannier-interpolation, as implemented in wannier90[74,75] and Phoebe[70], using a computationally converged 5x9x13 q-mesh for phonons and $25 \times 45 \times 65$ k-mesh for electrons. Bulk lattice thermal conductivity calculations were performed using Phoebe on a $5 \times 9 \times 13$ q-mesh (see Supplementary Section 4.1), accounting for phonon-phonon, phonon-isotope, and electron-phonon scattering. Phonon-disorder scattering was accounted for by postprocessing the Phoebe output linewidth. The Dirac delta appearing in the linewidth expression (see Supplementary Section 4.1) was broadened with a Gaussian smearing equal to $4.92 \times 10^{-5}$ eV.

The electronic band structure in Fig. 4e was computed using Quantum Espresso through the following standard procedure: (i) cell parameters and atomic positions of the 60-atom $Cu_3BHT$ structure were relaxed through density-functional theory calculations that computed the charge density self-consistently on a $2 \times 4 \times 4$ mesh of wavevectors uniformly spanning the Brillouin zone (as detailed above); (ii) after determining the relaxed cell, atomic positions, and charge density, the eigenvalues of the electronic Hamiltonian were computed over the path in the Brillouin zone connecting the special points Gamma, Y, C, Gamma, B with a dense discretization of 50 points per segment. Special points correspond to in-plane directions (Y, C) and an out-of-plane direction B.

## Simulation of GIWAXS pattern

2D-GIWAXS patterns were computed for the 2D van der Waals layered and the non-van der Waals layered $Cu_3BHT$ structures using the experimental lattices. The simulated pattern shown in Fig. 1b was generated by considering the zone axis [002] perpendicular to the substrate (i.e., face-on stacking orientation). The angular positions are obtained from the Materials Studio Reflex module. Pondered plane intensities are then computed using a home-made script to consider the amount of disorder in the films[76].

# Data availability

All the data supporting the findings and mentioned in this study are included in the Article and its Supplementary Information files. Raw data are available from the corresponding authors upon request for academic and non-commercial purposes. The DFT-optimized non-van der Waals layered $Cu_3BHT$ structure (optimized with PBEsol functional and Grimme-D2 vdW corrections) used for electronic band structure calculation and heat transport simulation is available on Materials Cloud Archive[77]: https://doi.org/10.24435/materialscloud:f3-28.

# Code availability

The Quantum Espresso software is available at https://www.quantum-espresso.org; the SSSP efficiency library is available at https://www.

materialscloud.org/discover/sssp/table/efficiency; the Hiphive package is available at https://gitlab.com/materials-modeling/hiphive; the Wannier90 package was included in the Quantum Espresso installation and it is also available at https://wannier.org/; the Phoebe software is available at https://github.com/phoebe-team/phoebe. The script used for GIWAXS pattern simulations was developed in-house at the Laboratory for Chemistry of Novel Materials in the University of Mons. It can be shared by the corresponding authors upon request for academic and non-commercial purposes.

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

## Acknowledgements

We gratefully acknowledge Zhichao Pan and Jin-Hu Dou (School of Materials Science and Engineering, Peking University) for providing a sample from which the newly reported Cu₃BHT structure was found and for providing the CIF file. H.-I.U. Thank William Wood (University of Cambridge) for discussion on Hall effect and Jonathan Rawle (Diamond Light Source Beamline I-07) for GIWAXS measurement assistance. H.-I.U., H.S., N.F., and H.N. acknowledge the support from the Engineering and Physical Sciences Research Council (EPSRC) and the Japanese Society for the Promotion of Science (JSPS) through a core-to-core grant (EP/S030662/1). H.S. thanks the Royal Society for a Research Professorship (RP/R1/201082). I.J. acknowledges funding from the European Research Council (Advanced Grant 101020872) and a Royal Society University Research Fellowship (URF\R1\231287). J.F.O. acknowledges funding from EPSRC Nano Doctoral Training Centre (EP/L015978/1). D.C. and D.B. acknowledge the support from the Energy Transition Fund of the Belgian Federal Government (FPS Economy) within the T-REX project. The computational resources for the Raman simulations were provided by the Consortium des "Equipements de

Calcul Intensif" (CÉCI) funded by the Belgian National Fund for Scientific Research (F.R.S.-FNRS) under Grant 2.5020.11. SED studies were supported by the access to e02 at ePSIC Diamond Light Source (MG32017). We thank the Diamond Light Source Beamline I-07 for beamtime (SI35227, SI35227–1). We also thank the EPSRC and the Henry Royce Institute for access to the thermoelectric test equipment (Cambridge Royce facilities grant EP/P024947/1 and Sir Henry Royce Institute—recurrent grant EP/R00661X/1). K.I. acknowledges support from the Winton and Cavendish Scholarship at the Department of Physics in the University of Cambridge. M.S. acknowledges support from Gonville and Caius College at the University of Cambridge. The computational resources for the first-principles simulations based on the Wigner transport equation were provided by: (i) the Kelvin2 HPC platform at the NI-HPC Centre (funded by EPSRC and jointly managed by Queen's University Belfast and Ulster University); (ii) the UK National Supercomputing Service ARCHER2, for which access was obtained via the UKCP consortium and funded by EPSRC (EP/X035891/1).

## Author contributions

H.-I.U. conceived the project, prepared all samples and devices, designed and conducted all experimental characterizations with assistance from some of other authors except for the SED characterization, which was done and analysed by J.F.O independently. H.-I.U. analysed data, and led the scientific development of this work with H.S. and M.S. together. I.E.J. greatly helped with GIWAXS measurement and initial data processing of the raw data. J.F.O. and I.E.J. reviewed and commented on the initial draft. N.F. synthesized the BHT molecule under H.N. supervision. D.C. performed the GIWAXS simulation under D.B. supervision. K.I. and M.S. performed the electronic band structure calculation and the theoretical analysis of the thermal conduction properties based on the Wigner Transport Equation, wrote the section of heat transport simulation in the paper, and contributed to the explanation of the corresponding experimental results. H.S. supervised the project. H.-I.U. wrote the first draft, and revised it with significant contribution from H.S. and inputs from M.S. All authors reviewed and commented on the manuscript.

## Competing interests

The authors declare no competing interests.
