## [Transparent Peer Review file · Nature Communications]

Defect-tolerant electron and defect-sensitive phonon transport in quasi-2D conjugated coordination polymers

Corresponding Author: Professor Henning Sirringhaus

This manuscript has been previously reviewed at another journal. This document only contains information relating to versions considered at *Nature Communications*. Mentions of prior referee reports have been redacted.

Version 0:

Reviewer comments:

Reviewer #1

(Remarks to the Author)

[Note from the editor: reviewer 1 also looked over the responses relevant to added calculations]

The manuscript presents the results of a study on the structure–thermoelectric property relationships in 2D conjugated MOF films of copper benzenehexathiol (Cu-BHT). The study finds that this material exhibits phonon-glass, electron-crystal transport behavior, with the highest electrical conductivities observed in disordered films with paracrystallinity exceeding 10%.

Temperature-dependent thermoelectric and magnetotransport measurements suggest a two-carrier, ambipolar transport regime, with contributions from electron and hole pockets in the band structure, explaining the electron-crystal behavior. Meanwhile, temperature-dependent diffraction and Raman spectroscopy confirm that the phonon-glass behavior results from small-amplitude, quasi-harmonic lattice dynamics induced by locally varying lattice strains.

These findings are significant and novel and address a major limitation in the literature regarding the wide variation in reported electrical conductivities and the lack of studies on thermal conduction. The methodology and data quality is good. I recommend publication of this article. The first principle calculations performed to address the nature of vibrational heat carriers in Cu-BHT films fully explain the low value of the thermal conductivity. The paper should be published now as is.

Reviewer #4

(Remarks to the Author)

[Note from the editor: Reviewer 4 was recruited to look over the comments from reviewer 2 and reviewer 3 from the previous round of review]

My overall assessment of the revised manuscript is, while I appreciate the extensive analysis of the CuBHT that the authors have performed, I have reservations about the overall impact of the study because of the reasons stated below and therefore would not recommend the work to be published in Nature Communications.

1) The defect tolerance for high electrical conductivity is an overstatement since any doped polymer semiconductor will have many chemical and structural defects (how to quantify and benchmark these is a completely different story) and still achieve high electrical and low thermal conductivity. While the authors have studied a relatively new class of materials, it doesn't set them apart from e.g. doped polymer semiconductors regarding thermoelectric performance.

2) It's intriguing that the presented material can combine low κ with high σ , but this has already been demonstrated for organic conductors such as PEDOT:PSS which offers higher Seebeck coefficients (ref 10), and therefore higher ZT. I feel that benchmarking new materials needs to consider the Seebeck coefficient as well instead of focusing on a favorable ratio of only two parameters that determine the thermoelectric performance.

3) The material appears to suffer from morphological defects, since the supposedly more crystalline and perfect materials show larger domains that also seem to have a different texture (orientation of domains with respect to one another) which then also explains their lower electrical conductivity.

4) I would argue that the developed material COULD represent innovation, but not with the current material characterization since it's unknown what exactly has been synthesized. TEM would be able to shed some light on the nanostructure and grain orientation that is obtained. I do acknowledge that structural analysis can be a difficult task since this is greatly facilitated by solubility. There is also no control over the defect densities, opposed to what the authors claim (rather, it's influencing).

Reviewer 2:

Comment 1: [REDACTED]

The response of the authors is valid, comparing the presented materials with semimetals is not correct, and should instead be compared to doped polymer semiconductors.

Comment 2: [REDACTED]

I agree with reviewer 2 here. The authors respond to this comment by relating the thermal conductivity to inorganic layered materials whereas they should compare the found thermal conductivities with organic materials (e.g. doped conjugated polymers), which are on the order of 0.1-1 W m⁻¹ K⁻¹ and approximately what the authors find.

Comment 3: [REDACTED]

Considering the way the material has been synthesized -an interfacial reaction that must involve the diffusion of at least 1 species (likely the BHT into the water phase rather than the copper acetate in the organic phase) to obtain the several 100 nm thick layers that were reported -and the fact that structural analysis shows anisotropy, I would have to agree with reviewer 2. The material appears to carry both structural and morphological defects which would offer a good explanation for the low thermal conductivity (not even considering the organic nature of the material developed). The authors have attempted to address the reviewers concerns by carrying out modeling and adding to the manuscript, I do not have the knowledge on computational chemistry to make a fair assessment of the contribution.

Comment 4: [REDACTED]

I would argue that the developed material COULD represent innovation, but not with the current material characterization since it's unknown what exactly has been synthesized. TEM would be able to shed some light on the nanostructure and grain orientation that is obtained. I do acknowledge that structural analysis can be a difficult task since this is greatly facilitated by solubility. There is also no control over the defect densities, opposed to what the authors claim (rather, it's influencing). It's intriguing that the presented material can combine low κ with high σ , but this has already been demonstrated for organic conductors such as PEDOT:PSS which offers higher Seebeck coefficients (ref 10), and therefore higher ZT. I feel that benchmarking new materials needs to take into account the Seebeck coefficient as well instead of focusing on a favorable ratio of two parameters. I don't think the authors have adequately addressed this concern.

Reviewer 3

Comment 1: [REDACTED]

I think the first comment should be a bit more nuanced since the authors clearly did more than increase the electrical conductivity of cMOF's. The claim though that that the defect tolerance for high electrical conductivity is surprising (conclusion) is an overstatement since any doped polymer semiconductor will have many chemical and structural defects (how to quantify and benchmark these is a completely different story) and still achieve high electrical and low thermal conductivity.

Comment 2: [REDACTED]

This is also a valid comment, and I doubt whether the ZT will actually outperform doped organic materials since they typically have higher Seebeck coefficients. This comment is in line with R2's, 4th comment, and therefore not adequately addressed.

Comment 3 [REDACTED]

I think the authors are correct here in their response, the model used should be different.

Comment 4: [REDACTED]

I agree with reviewer 3 here.

The response from the authors is not convincing since the supposedly more crystalline and perfect materials show larger domains that also seem to have a different texture (orientation of domains with respect to one another) which then also explains their lower electrical conductivity.

Version 1:

Reviewer comments:

Reviewer #4

(Remarks to the Author)

Dear authors,

I have reviewed your comments and I believe they are satisfactory addressed. I am pleased by the added data and discussion to put the presented material in context. Therefore, I recommend acceptance of your manuscript.

We are grateful for the helpful and constructive comments by the referees and have carefully revised the manuscript according to their suggestions.

In the detailed replies below the reviewers' comments are in **black**, and our responses are in **blue**. Texts in the original manuscript are in **green** and the corresponding revisions taken in the main text and supporting information are in **orange**.

Reviewer #1 (Remarks to the Author):

[Note from the editor: reviewer 1 also looked over the responses relevant to added calculations]

The manuscript presents the results of a study on the structure–thermoelectric property relationships in 2D conjugated MOF films of copper benzenehexathiol (Cu-BHT). The study finds that this material exhibits phonon-glass, electron-crystal transport behavior, with the highest electrical conductivities observed in disordered films with paracrystallinity exceeding 10%.

Temperature-dependent thermoelectric and magnetotransport measurements suggest a two-carrier, ambipolar transport regime, with contributions from electron and hole pockets in the band structure, explaining the electron-crystal behavior. Meanwhile, temperature-dependent diffraction and Raman spectroscopy confirm that the phonon-glass behavior results from small-amplitude, quasi-harmonic lattice dynamics induced by locally varying lattice strains.

These findings are significant and novel and address a major limitation in the literature regarding the wide variation in reported electrical conductivities and the lack of studies on thermal conduction. The methodology and data quality is good. I recommend publication of this article. The first principle calculations performed to address the nature of vibrational heat carriers in Cu-BHT films fully explain the low value of the thermal conductivity. The paper should be published now as is.

Response: We thank reviewer 1 for the time to review this manuscript again, and we are grateful for providing support and constructive feedback to our work.

Reviewer #4 (Remarks to the Author):

[Note from the editor: Reviewer 4 was recruited to look over the comments from reviewer 2 and reviewer 3 from the previous round of review]

My overall assessment of the revised manuscript is, while I appreciate the extensive analysis of the CuBHT that the authors have performed, I have reservations about the overall impact of the study because of the reasons stated below and therefore would not recommend the work to be published in Nature Communications.

1) The defect tolerance for high electrical conductivity is an overstatement since any doped polymer semiconductor will have many chemical and structural defects (how to quantify and benchmark these is a completely different story) and still achieve high electrical and low thermal conductivity. While the authors have studied a relatively new class of materials, it doesn't set them apart from e.g. doped polymer semiconductors regarding thermoelectric performance.

Response: We thank reviewer 4 for the comments. The concern on the defect tolerance motivates us to systemically and carefully compare the electrical properties of doped polymer semiconductors and Cu-BHT as a function of ordering. It is well known that the level of ordering/defect of materials can be quantified by paracrystallinity (g)^{1,2}. The higher paracrystallinity the higher defective level. We find that commonly *doped polymer semiconductors are not defect-tolerant but Cu-BHT is* because of two reasons:

- i. The correlation between electrical conductivity and paracrystallinity is negative for doped polymer semiconductors but insensitive (or even mildly positive) for Cu-BHT. Jacobs et al.³ systemically studied the effects of ordering on the electrical conductivities of doped polymer semiconductors and clearly showed a negative correlation for the representative high-performing polymers PBTTT (**Figure R1a**), P3HT (**Figure R1b**), and DPP-BTz (**Figure R1c**). The negative correlation indicated that the electrical conductivities of doped polymers are not defect-tolerant, but defect-suppressed. Importantly, however, **Figure R1d** clearly shows that at similar ordering/defective levels, particularly when $g > 10\%$, Cu-BHT has significantly higher electrical conductivities than these conducting polymers, and also exhibits a much weaker, and even mildly positive correlation with paracrystallinity.
- ii. Electrical conductivities remain thermally activated for most doped polymer semiconductors, but truly metallic for Cu-BHT films of Cu/BHT ratios 5 and 5.5 with $g > 10\%$, more specifically, mildly reduced for PBTTT even with conductivity $> 3000 \text{ S cm}^{-1}$ (**Figure R1e**)⁴, and dropped by four orders of magnitude for DPP-BTz and IDTBT upon cooling (**Figure R1f,g**)⁴, in significant contrast to the metallic temperature dependence (increase in conductivity upon cooling) of Cu-BHT (**Figure R1h**). In a conducting polymer it is extremely difficult to observe a temperature dependent conductivity that remains metallic ($d\sigma/dT < 0$) down to temperatures below 10K^{5,6}.

We therefore argue, that doped polymer semiconductors are NOT really defect-tolerant in electrical conduction and, generally speaking, (at similar doping levels) in conducting polymer the electrical conductivity is the higher the higher the structural ordering; this is in contrast to Cu-BHT that is defect-tolerant. Such a defect-tolerant electrical transport mechanism and the more outstanding electrical conductivity values at similar ordering/defective levels set Cu-BHT apart from doped polymer semiconductors.

Additionally, in the thermal conduction, in fundamental aspect, very different microscopic or molecular origins can give rise to similar macroscopic conduction phenomena. However, it is important to understand and differentiate the fundamental reasons for the low thermal conductivities of doped polymer semiconductors and of Cu-BHT. At least for Cu-BHT, our study presents the first quantitatively accurate *ab-initio* theoretical investigation that distinguishes the roles of (i) intrinsic phonon-phonon scattering, (ii) electron-phonon scattering, and (iii) structural disorder in thermal conduction. Our results reveal that *the low thermal conductivity arises not only from the structural disorder, but also has important contributions from the heavy nature of Cu atoms and the inactive vibrations of C atoms that hinder the development of heat transfer*, and thus giving exceptionally low lattice thermal conductivities of $\sim 1 \text{ W m}^{-1} \text{ K}^{-1}$, even in perfect crystal (simulation) and $\sim 0.2 \text{ W m}^{-1} \text{ K}^{-1}$ in films (simulation that accounts for structural disorder and experiment). This is fundamentally different from the intrinsic reason of the soft nature of polymer semiconductors that leads to significant anharmonic effect, which contribute to low thermal conductivities. Cu-BHT is NOT as soft as a conjugated polymer and exhibits mainly harmonic lattice dynamics.

Figure R1. Electrical properties of doped polymer semiconductors and Cu-BHT as a function of paracrystallinity, i.e. ordering/defect. a)-c) Electrical conductivities as a function

of paracrystallinity for PBTTT, P3HT, and DPP-BTz. **d)** Summary of the electrical conductivities of some typical, representative doped polymer semiconductors and Cu-BHT films, clearly showing two different regions. **e)-h)** Temperature dependence of electrical conductivity for doped PBTTT, DPP-BTz, and IDTBT films and Cu-BHT films studied in this work. The data of doped polymer semiconductors in this Figure is reproduced from literature^{3,4}.

In conclusion, although Cu-BHT and doped polymer semiconductors have similarly low thermal conductivity, there are important differences, which we believe are important to investigate in the present work: The electrical conductivity of Cu-BHT is defect-tolerant and higher at similar levels of structural disorder/paracrystallinity than in a conducting polymer, while that of doped polymer semiconductors is defect-suppressed. The charge and heat transport mechanisms of Cu-BHT are different from doped polymer semiconductors', setting Cu-BHT apart from doped polymer semiconductors in both performance and fundamental aspects. Our work also provides a sophisticated computational framework for studying the microscopic processes responsible for heat conduction in Cu-BHT that could, in principle, be applicable to conducting polymers with different atomic and electronic structures, to understand their heat conduction physics in more details.

To clarify the point of defect tolerance that reviewer 4 concerned, we have now particularly added **Figure 5a** and corresponding discussion "This defect tolerance is unprecedented and surprising, setting it apart from polymer (semi)conductors in which electron transport is defect-suppressed (Fig. 5a) and in which the electrical conductivity typically exhibits a non-metallic temperature dependence ($d\sigma/dT > 0$) at low temperatures." in the Main Text, page 13.

Figure 5. Comparison of electrical conductivity σ , paracrystallinity g , and lattice thermal conductivity κ_{latt} of different types of materials at room temperature. a, Comparison of the dependence of σ on g in Cu-BHT and various conducting polymers from the literature. Cu-BHT exhibits significantly higher electrical conductivity at a given paracrystallinity value than conducting polymers. Data for doped polymer semiconductors are extracted from reference³. **b,** Plot of σ vs κ_{latt} . These materials include organic materials, inorganic materials, commercial Bi_2Te_3 alloy, cage-like compounds such as skutterudites, open frameworks such as porous MOFs, and non-open, non-porous Cu-BHT (this work and previous work). The values for our

Cu-BHT, indicated by stars, highlight the new, advantageous defect-tolerant charge and defect-sensitive phonon transport regime discovered in this work that provides a remarkable electrical conductivity/lattice thermal conductivity ratio $\sigma/\kappa_{\text{latt}}$ of up to $60 \times 10^4 \text{ S K W}^{-1}$ that is 5 – 16 times higher than that of other state-of-the-art thermoelectric materials. The figure does not include materials in which thermoelectric enhancement is attributed to using special techniques such as nanostructuring or alignment. Detailed data from the relevant references are provided in Supplementary Table 3.

2) It's intriguing that the presented material can combine low κ with high σ , but this has already been demonstrated for organic conductors such as PEDOT:PSS which offers higher Seebeck coefficients (ref 10), and therefore higher ZT. I feel that benchmarking new materials needs to consider the Seebeck coefficient as well instead of focusing on a favorable ratio of only two parameters that determine the thermoelectric performance.

Response: We thank for reviewer's comments, but argue that it is not reasonable to judge the scientific quality and potential impact of our work primarily on whether, we can reach competitive thermoelectric performance with conducting polymers, such as PEDOT:PSS. This is for the following reasons:

- i) Thermoelectrics are not the only application of materials with high electrical defect tolerance and ultralow thermal conductivity. Our materials and fundamental insights could also be used in applications beyond thermoelectrics that require multi-functional integration of electrical conductors into packages that also provide thermal insulation, for example as required in thermionic devices and some wearable electronic applications, integration of conducting sensors into building insulation or some aerospace and energy applications⁷⁻¹¹. Our work is not primarily application focussed, but we believe that in light of these applications the interplay between heat and charge transport, which in our Cu-BHT exhibit a negative correlation, is sufficiently unusual and interesting, that it is worth understanding at a fundamental level.
- ii) The thermoelectric properties of PEDOT:PSS have been widely studied by many groups over more than four decades since PEDOT was first reported by Bayer in 1988 and significant progress has been made in the thermoelectric performance of PEDOT:PSS over the decades. In particular, replacing the counterion PSS with tosylate ions (Tos) and controlling/reducing the doping level by treatments with a reducing agent (TDAE) enabled achieving a higher Seebeck coefficient and higher thermoelectric performance with ZT of 0.25¹² (i.e. the ref 10 in the Main Text mentioned by reviewer 4). We do appreciate and gratefully acknowledge these important advances in organic thermoelectrics and the relevant state-of-the-art performance. However, we also note that in many regular PEDOT:PSS formulations the Seebeck coefficient is not very high and typically on the order of only 10 – 20 $\mu\text{V/K}$. In fact, we find that both the ZT values and power factor σS^2 of our Cu-BHT films ($ZT = 0.0015$, $\sigma S^2 = 5.0 \mu\text{W m}^{-1} \text{K}^{-2}$ for Cu/BHT = 2) are in the same range of those of many pristine PEDOT:PSS films and PEDOT:PSS films treated with EG, DMSO, H_2SO_4 , and NaOH ($ZT = 0.0004 - 0.005$, $\sigma S^2 = 0.03 - 12.0 \mu\text{W m}^{-1} \text{K}^{-2}$)¹³⁻¹⁵, as shown in **Figure R2a**. This shows that

the thermoelectric performance of our Cu-BHT, though not record breaking, is at least credible.

It is entirely possible, that at some point in the future, just like in PEDOT, methods will be discovered to enhance the Seebeck coefficient of Cu-BHT to values that will then enable state-of-the-art thermoelectric performance. The insights into the transport physics of Cu-BHT gained in our work, in particular, the demonstration that both electron and hole-dominated bands contribute to conduction, which lowers the achievable Seebeck coefficient, do in fact inform concrete strategies that could be used in the future to achieve this, such as tuning the Fermi level within the band structure by (de)doping to a position, where transport is more unipolar. We are currently working on this actively using techniques such as electrochemical gating.

We believe that our work has significant value in relation to future thermoelectric applications of Cu-BHT, in that it identifies the favourable defect-tolerant electrical transport and defect-sensitive heat transport regime, that is achievable in these materials and is in fact the most important requirement for any promising thermoelectric materials, AND the fundamental cause of the low Seebeck coefficient currently observed in these materials and potential strategies for enhancing the Seebeck coefficient.

Figure R2. Comparison of the thermoelectric performances of PEDOT and Cu-BHT and their development progress. a) ZT values and b) power factor σS^2 of PEDOT:Tos, PEDOT:PSS, and Cu-BHT. c) The number of publications per year for PEDOT and Cu-BHT, where the PEDOT means including all counterions (PSS, Tos, etc.).

As suggested by the reviewer 4, to take the Seebeck coefficient into account, and clarify it is comparable to PEDOT:PSS, we have now rephrased the concluding section of the paper on page 14 to reflect this: *At present the thermoelectric performance of our Cu-BHT is credible, but not particularly high with $ZT = 0.0015$ and power factor $\sigma S^2 = 5.0 \mu\text{W m}^{-1} \text{K}^{-2}$; this is comparable to that of many conducting polymers. However, our findings suggest that*

techniques to tune the Fermi level into an effective, one-band regime and control the unipolar carrier concentration could be effective for enhancing the Seebeck coefficient and thus thermoelectric performance. This may be regarded as analogous to the development of polymer thermoelectrics, where enhancing thermoelectric performance also required advanced (de)doping methods to boost the Seebeck coefficient¹⁰. In any case, conjugated coordination nanosheets could offer attractive properties, such as ease of processing and defect tolerance, for applications, that require fast charge, but slow heat transport, which are not limited to thermoelectrics, but also include multi-functional thermal barrier coatings, as required in thermionic devices, some wearable electronic applications, integration of conducting sensors into building insulation or aerospace and energy applications⁷⁻¹¹.

3) The material appears to suffer from morphological defects, since the supposedly more crystalline and perfect materials show larger domains that also seem to have a different texture (orientation of domains with respect to one another) which then also explains their lower electrical conductivity.

Response: We thank reviewer 4 for commenting on this. We had actually performed GIWAXS and SED measurements to identify the domains orientation among different compositions in our original manuscript. The results clearly showed that even though the films can have different defect levels and different morphologies, their orientations of domains are always the same and mainly face-on. Particularly, by acquiring and indexing each diffraction pattern in the scanned area in SED measurements, an orientation map (i.e. **Figure 1h,i**) can be computed. These maps directly visualise the distribution of grain orientation that fully explains reviewer 4's concern. The red regions indicate domains that are oriented face-on, while black regions are regions in which the structural order is insufficient to assign a grain orientation. The maps show clearly that face-on oriented grains remain dominant in all films and there is no difference in texture.

In the original Main Text, page 3 “Scanning electron diffraction (SED) in transmission mode was employed to directly visualize the stacking orientation on the nanoscale (Fig. 1h,i, Supplementary Fig. 11). A face-on stacking preference with the [002] zone axis normal to substrate (indicated in red), is seen to dominate in both Cu/BHT ratios of 2 and 3.5, while edge-on preference, i.e. zone axes [200] and [020] normal to substrate (green and blue, respectively), are less prominent.”

Figure 1h,i, SED maps visualizing the stacking orientation of films with Cu/BHT ratio of 2 and ratio of 3.5 on the nanoscale. Face-on dominant grains (zone axis [002] mostly normal to the substrate) are highlighted in red, edge-on dominant grains (zone axes [200] and [020] mostly normal to the substrate) are highlighted in green and in blue, respectively. White streak lines correspond to grains with intermediate/mixed orientation. Black areas show the regions where diffraction is not good enough to estimate the zone axes.

To be clearer and more accurate we have now also modify some of the discussion of GIWAXS on page 3 of the Main Text to be “The most crystalline films exhibit a pronounced face-on-preferred orientation, as evident by the (002) observed in q_z direction, and films grown with higher Cu/BHT ratio are also mainly face-on, but exhibit larger regions with a more amorphous microstructure in between the face-on domains (Fig. 1b-d, Supplementary Fig. 10b-d)”

Also, in the original Supplementary Information, page 14, **Figures S10e,f**, which are the diffraction linecut profiles along q_{xy} and q_z extracted from the GIWAXS images **Figure S10a-d**, clearly shows that diffraction peak (002) is along the q_z direction while peaks (200), (400), and (600) are along the q_{xy} direction, clearly indicating the layered structure of the Cu-BHT films is parallel to the substrate, i.e. face-on orientation. To ensure that these important details are not overlooked we have added the following clarifying statement to Supplementary Information page 14: “..., clearly showing that diffraction peak (002) is along the q_z direction while peaks (200), (400), and (600) are along the q_{xy} direction, so that the layered structure of the Cu-BHT films is identified to be parallel to the substrate, i.e. face-on orientation.”

4) I would argue that the developed material COULD represent innovation, but not with the current material characterization since it's unknown what exactly has been synthesized. TEM would be able to shed some light on the nanostructure and grain orientation that is obtained. I do acknowledge that structural analysis can be a difficult task since this is greatly facilitated by solubility. There is also no control over the defect densities, opposed to what the authors claim (rather, it's influencing).

Response: We appreciate reviewer 4 for supporting that this material could represent innovation. Our reference to being able to control defect densities referred to varying the molar ratio between the reactants Cu^{2+} and BHT in a systematic manner and being able to vary the paracrystallinity and degree of structural disorder in this way. We show that it is possible to link the molar ratio between Cu^{2+} and BHT added into the growth solutions to the actual atomic ratio of Cu to S present in the films, providing some insights for future defect engineering/control in conducting MOFs. This enabled us to study the impact of structural disorder on charge and heat conduction in this class of materials systematically for the first time. Specifically, our work addresses the question behind the unknown, huge electrical conductivity variation spanning over 0 – 2500 S cm^{-1} that has been observed in Cu-BHT films in several studies^{16–19} using the same synthesis method (liquid-liquid interfacial synthesis). ON balance, we acknowledge that the work “control” might be too strong and have replaced it with “influence” instead, as suggested by the referee.

Corresponding chemical composition had been characterised by EDX in our original manuscript. To be clearer for readers, we have now further clarified the defect density by revising related discussion in the Main Text page 3 to be: “EDX was performed to quantify the chemical composition and defect density (Supplementary Fig. 4). The actual Cu-to-S atomic ratio is positively correlated with the Cu/BHT molar ratio used during film growth (Fig. 1f). Under all growth conditions used the composition of the films remains Cu-rich, as evident by the Cu/S atomic ratio higher than the ideal value of 0.5, even for an excess of organic ligand BHT was added. These results suggest that the majority of chemical defects in our films are BHT vacancies and the densities are estimated to be one BHT vacancy in every 3, 2, 1.8, and 1.4 unit cells in the films prepared by Cu/BHT ratios of 2, 3.5, 5, and 6.5 during the synthesis, respectively.”

Regarding material characterisations, we agree with reviewer 4 that TEM would be helpful to shed some light on the nanostructure, and we have attempted to perform TEM measurements. However, as the reviewer acknowledged, structural analysis can be a difficult task and, unfortunately, we failed in resolving the nanostructure by TEM lattice imaging. High-resolution TEM measurement requires the samples to be very thin. We have been struggling with preparing very thin Cu-BHT films on TEM grids, because transferring thin films from liquid-liquid interface to TEM grids is challenging. Importantly, these thin films (thickness < 20 – 30 nm) become very fragile and usually shatter on the TEM grids, presumably due to the change of surface tension during drying process. There were some squares that were still intact (**Figure R4a**), but the samples on there were usually too thick (that’s why these regions were not so fragile and could intact) and did not work for high-resolution TEM measurements (**Figure R4b-f**).

Figure R4. TEM measurements. **a**, A representative optical image of the films that were still intact on TEM grids. **b-f**, TEM images of the regions having the films still intact, but the films on there were usually too thick (so that not fragile).

In the absence of TEM imaging, we have made every effort to characterise our films and have used a wide range of techniques, including EDX, GIWAXS, SED, Raman spectroscopy, and XPS, combined with first-principles simulations, to identify that what we synthesized is the non-van der Waals layered $\text{Cu}_3\text{BHT}^{20}$ films with different defect levels.

Particularly, all diffraction peaks seen in the GIWAXS experiments can be assigned, well matched with the first GIWAXS simulation of the non-van der Waals layered Cu_3BHT we performed in this work, as shown in **Figure R5**. More detailed GIWAXS comparison and analysis had been included in **Figures S5-S7** and discussed on pages 9, 10 in our original Supplementary Information.

Figure R5. GIWAXS experiment and simulation. **a**, Simulated GIWAXS pattern for non-van der Waals layered structure Cu₃BHT. **b**, Experimental GIWAXS pattern of our films studied in this work. **c**, The origins of some representative diffraction peaks.

We have now also performed additional electron diffraction characterisation by SED on the films of Cu/BHT ratio 3. As shown in **Figure S12**, the electron diffraction pattern in the red box shows the zone axis [00 \bar{l}] vector with diffraction peak (002) found at $0.275 \pm 0.015 \text{ \AA}^{-1}$ while that the green box exhibits the zone axis [h00] with (200) found at $0.126 \pm 0.015 \text{ \AA}^{-1}$. This kind of electron diffraction has been performed at every pixel of different scan areas during the SED measurements. By acquiring and indexing each diffraction pattern in the scanned area, this is how the grains orientation maps (i.e. **Figure 1h,i**, see our response to reviewer 4’s comment 3 for more details) were collected. Some representative diffraction images showing the in-plane lattice are now further added as **Figure S13** as examples. These lattice spacings for Cu/BHT ratio 3.5 are the same as those of Cu/BHT ratio 2 (**Figure S11**) that we had included in our original Supplementary Information page 15 and once again match with the non-van der Waals layered Cu₃BHT²⁰. Below **Figures S12,S13** along with text “Similarly, same results were found for Cu/BHT ratio 3.5 (**Figure S12**)... Some representative diffraction images showing the in-plane lattice are further added as **Figure S13** as examples. These lattice spacings match with the in-plane spacing of the non-van der Waals layered” have now been added into our Supplementary Information pages 15, 16.

XPS experiment has now also been carried out on different Cu-BHT films and added into Supplementary Information page 17 as **Figures S14,S15** with text “XPS experiment was performed on Cu-BHT films prepared by different Cu/BHT ratios and they all show spectra essentially the same as what the literature reported for Cu₃BHT²⁰ (**Figures S14,S15**).”

Figure S12. SED experiment on films of Cu/BHT ratio 3.5. **a**, vDF images taken by SED. **b,c**, Representative edge-on and face-on electron diffraction patterns taken at the red and the green boxes labelled in **a**. **d,e**, Diffraction profiles in reciprocal spacing correspond to the lines in **b** and **c** respectively.

Figure S13. Electron diffraction images from SED measurements of films of Cu/BHT ratio 3.5. **a, b, c** are different regions. They all show a lattice matched with the in-plane lattice of the non-van der Waals layered Cu_3BHT .

Figure S14. XPS spectra of Cu-BHT films prepared by different Cu/BHT ratios. **a**, Cu/BHT ratio 2. **b**, Cu/BHT ratio 3.5. **c**, Cu/BHT ratio 5.

Figure S15. High-resolution XPS Cu 2p, S2p, and C1s spectra of Cu-BHT films prepared by different Cu/BHT ratios. **a**, Cu/BHT ratio 2. **b**, Cu/BHT ratio 3.5. **c**, Cu/BHT ratio 5.

Reviewer 2:

Comment 1: [REDACTED]

Further comment from reviewer 4: The response of the authors is valid, comparing the presented materials with semimetals is not correct, and should instead be compared to doped polymer semiconductors.

Response: We thank reviewer 4 for commenting on this, which is in line with reviewer 4's comments 1) and 2). The main difference that set Cu-BHT apart from doped polymer semiconductors is that their transport mechanisms are fundamentally different and it gives Cu-BHT higher electrical conductivity than doped polymer semiconductors at comparable levels of structural disorder. As clarified in the response to reviewer 4's comment 1), we argue that Cu-BHT is defect-tolerant, but doped polymer semiconductors are not and such a defect tolerance property allows Cu-BHT to achieve several-time or one-order-of-magnitude higher

electrical conductivity than typical doped polymer semiconductors at the same level of structural disorder/paracrystallinity (**Figure R1d**). The difference in transport physics also manifests itself in their temperature dependences of electrical conductivities, that show thermally activated dependence with conductivities dropping by four orders of magnitude upon cooling in doped polymer semiconductors with paracrystallinity > 10% but metallic temperature dependence in Cu-BHT, also with paracrystallinity > 10%. This clear defect-tolerant property represents a new, advantageous thermoelectric transport region that suggests this class of new materials would be interesting and potentially promising in thermoelectric applications. Detailed comparison regarding their changes in electrical properties as a function of defect levels can be found in the response to reviewer 4's comment 1).

In terms of the comparison with conducting polymers we have addressed this above in relation to point 2 raised by reviewer 4. We argue that (i) thermoelectric applications are not the only application that require the unusual combination of high electrical conductivity and low thermal conductivity, and (ii) that conducting polymers have been studied for much longer and with higher efforts and that specific advanced (de)doping methods were needed to enhance the Seebeck coefficient to achieve high levels of thermoelectric performance. The thermoelectric performance (ZT and power factor) of many PEDOT and other conducting polymers is in fact very comparable to what we have achieved in Cu-BHT.

Comment 2: [*REDACTED*]

Further comment from reviewer 4: I agree with reviewer 2 here. The authors respond to this comment by relating the thermal conductivity to inorganic layered materials whereas they should compare the found thermal conductivities with organic materials (e.g. doped conjugated polymers), which are on the order of $0.1-1 \text{ W m}^{-1} \text{ K}^{-1}$ and approximately what the authors find.

Response: Thank reviewer 4 for commenting on this. We think the important point here is not just the low thermal conductivity, but that it is achieved simultaneously with a high and defect-tolerant electrical conductivity, providing an ideal mix of antithetical exceptionally low lattice thermal conductivities of $0.2 \text{ W m}^{-1} \text{ K}^{-1}$ below Kittel's limit and high electrical conductivities up to 2000 S cm^{-1} . The defect tolerant electrical conductivity, which as we argue above is in contrast to the defect-sensitive electrical conductivity of conducting polymers, is likely to reflect the two-dimensional nature of conjugation that is present in these 2D cMOFs. It is by no means obvious from the literature on 1D conjugated polymers that 2D cMOFs should have as low thermal conductivity as conducting polymers, particularly since other 2D electronic materials have of course exceptionally **high** thermal conductivity. As explained in more detail above our theoretical simulations identify clearly the fundamental reasons why this is possible. It is due to a combination of intrinsically low thermal conductivity due to the influence of the heavy Cu transition metal atoms, that could be achieved even in a single crystal, and advantageous defects that lower the thermal conductivity, but do not strongly affect the

electrical conductivity. We believe this insight is novel and constitutes a significant advance in the scientific understanding of these emerging materials.

Comment 3: [REDACTED]

Further comment from reviewer 4: Considering the way the material has been synthesized -an interfacial reaction that must involve the diffusion of at least 1 species (likely the BHT into the water phase rather than the copper acetate in the organic phase) to obtain the several 100 nm thick layers that were reported -and the fact that structural analysis shows anisotropy, I would have to agree with reviewer 2. The material appears to carry both structural and morphological defects which would offer a good explanation for the low thermal conductivity (not even considering the organic nature of the material developed). The authors have attempted to address the reviewers concerns by carrying out modeling and adding to the manuscript, I do not have the knowledge on computational chemistry to make a fair assessment of the contribution.

Response: We thank the reviewer 4 for this comments, which prompts us to explain more clearly the difference between the established concept of PGEC, and the concept of defect-tolerant charge transport and defect-sensitive heat transport discussed in the manuscript. In particular, the analysis we performed shows that Cu-BHT is not an intrinsic phonon glass, but a material displaying defect-tolerant charge transport and defect-sensitive heat transport, which is a behavior/mechanism different from PGEC. The reviewer's comment motivated us to highlight in the manuscript that PGEC is not the only mechanism that promotes high thermoelectric performance, but defect-tolerant charge transport and defect-sensitive heat transport can also drive high thermoelectric performance.

This comment also motivates us to highlight the computational advances carried out in the manuscript. The modelling we performed solves the problem from first principles, the Wigner transport equation accounting for anharmonic phonon-phonon scattering²¹ and, most importantly, electron-phonon scattering. This simulation is state-of-the-art for two reasons:

- i. It is the first ab-initio simulation of thermal transport in the recently reported non-van der Waals layered Cu-BHT structure²⁰ that accounts for phonon propagation and tunnelling using the recently developed Wigner formulation;

- ii. The linear-response solution of the Wigner formulation accounts not only anharmonic phonon-phonon scattering, as in the literature²¹, but includes — for the first time — electron-phonon scattering. This statement of novelty is supported by a recent work, which applied the Wigner transport equation to organic crystals²² without accounting for electron-phonon interactions. The present analysis for Cu-BHT includes such interactions and therefore represents a significant advance over the very recent work²².

The developments performed in this work will pave the way for the first-principles modelling of transport in semiconducting and metallic organic materials.

Comment 4: [REDACTED]

Further comment from reviewer 4: I would argue that the developed material COULD represent innovation, but not with the current material characterization since it's unknown what exactly has been synthesized. TEM would be able to shed some light on the nanostructure and grain orientation that is obtained. I do acknowledge that structural analysis can be a difficult task since this is greatly facilitated by solubility. There is also no control over the defect densities, opposed to what the authors claim (rather, it's influencing). It's intriguing that the presented material can combine low κ with high σ , but this has already been demonstrated for organic conductors such as PEDOT:PSS which offers higher Seebeck coefficients (ref 10), and therefore higher ZT. I feel that benchmarking new materials needs to take into account the Seebeck coefficient as well instead of focusing on a favorable ratio of two parameters. I don't think the authors have adequately addressed this concern.

Response: We thank reviewer 4 for further commenting on this, which is a combined comment of the structural characterisations and the comparison with doped organic semiconductors in particular PEDOT:PSS that are already raised and same as reviewer 4's comments 2) and 4) respectively. Our answer can be found there.

Reviewer 3

Comment 1: [REDACTED]

Further comment from reviewer 4: I think the first comment should be a bit more nuanced since the authors clearly did more than increase the electrical conductivity of cMOF's. The claim though that that the defect tolerance for high electrical conductivity is surprising (conclusion) is an overstatement since any doped polymer semiconductor will have many chemical and structural defects (how to quantify and benchmark these is a completely different story) and still achieve high electrical and low thermal conductivity.

Response: We thank reviewer 4 for further commenting on this. The concern of defect tolerance is identical to reviewer 4 comment 1), our response can be found there.

Comment 2: [REDACTED]

Further comment from reviewer 4: This is also a valid comment, and I doubt whether the ZT will actually outperform doped organic materials since they typically have higher Seebeck coefficients. This comment is in line with R2's, 4th comment, and therefore not adequately addressed.

Response: We thank reviewer 4 for encouraging the comparison of the thermoelectric performance of Cu-BHT with that of doped organic semiconductors again here. As argued in more detail above, at the moment, the ZT of Cu-BHT is at the same level as that of many conducting polymers, including many formulations of PEDOT:PSS. Notably, Cu-BHT is very much more a recent and relatively unexplored discovery than PEDOT:PSS and is far from being fully optimised in its thermoelectric development history. It is entirely possible, that through the use of more controlled doping methods higher Seebeck coefficients and state-of-the-art thermoelectric performance will be achievable. We argue that the work reported in the present work lays the scientific foundations that justifies looking at the thermoelectric properties of Cu-BHT and other cMOFs more seriously. A more detailed reply can be found above in response to reviewer 4's comment 2).

Comment 3: [REDACTED]

Further comment from reviewer 4: I think the authors are correct here in their response, the model used should be different.

Response: We are very grateful that reviewer 4 supports us here.

Comment 4: [REDACTED]

Further comment from reviewer 4: I agree with reviewer 3 here. The response from the authors is not convincing since the supposedly more crystalline and perfect materials show larger domains that also seem to have a different texture (orientation of domains with respect to one another) which then also explains their lower electrical conductivity.

Response: We thank reviewer 4 for further commenting on this. As discussed above, our experimental results disagree that the defect-insensitive electrical conductivity comes from either different texture or large boundaries.

Even though the films with different compositions have different morphologies, both GIWAXS and SED results clearly revealed that the domains orientation among different compositions is the same and mainly face-on, i.e. with the layered structure parallel to the substrate. This is evidenced in **Figure 1 h,i** by the governing of zone axis [002] normal to substrate (indicated in red) in the SED mapping of both Cu/BHT ratios of 2 and 3.5 and in **Figure 1b-d, Figure S10e,f** by the diffraction peaks (002) appearing along the q_z direction and (200), (400), and (600) appearing along the q_{xy} direction in the GIWAXS images for all compositions measured, indicative of a mainly face-on orientation regardless of defect levels and morphologies. The grain orientation concern here is same as the comment 3) of reviewer 4. Detailed response to the grain orientation also can be found there.

The defect-insensitive electrical conductivity also cannot be interpreted by the grains being larger than electrical energy exchange length scales. If this were the true reason, we would have observed a decrease in electrical conductivities with increasing defect levels and decreasing

grain sizes. However, this did not happen. What experiments show is that in contrast, the electrical conductivity increased with increasing defect levels and decreasing grain sizes. The most crystalline, more chemical and structural perfect films with larger X-ray coherence length of 18.5 nm (i.e. grain size, and, notably not as large as ~100 nm considered by Reviewer 3) showed lower (~600 S/cm) and thermally activated electrical conductivity while the non-crystalline, more defective films with smaller X-ray coherence length of 9.4 nm (i.e. grain size) showed higher (~1500 S/cm), metallic temperature-dependent electrical conductivity. This is what we discovered and discussed in this work by “defect-tolerant charge transport” in the manuscript. Corresponding data had been summarised in **Figures 1, 2**.

Figure 1 c, d, Experimental GIWAXS and SEM images for a crystalline sample with Cu/BHT ratio of 2 and a more amorphous sample with a Cu/BHT ratio of 5, respectively. **g**, X-ray coherence length and paracrystalline disorder evaluated from WA analysis for the in-plane ($h00$) and out-of-plane ($00l$) diffractions as a function of Cu/BHT ratio.

References in this response letter

1. Prosa, T. J., Moulton, J., Heeger, A. J. & Winokur, M. J. Diffraction line-shape analysis of poly(3-dodecylthiophene): a study of layer disorder through the liquid crystalline polymer transition. *Macromolecules* **32**, 4000–4009 (1999).
2. Rivnay, J., Noriega, R., Kline, R. J., Salleo, A. & Toney, M. F. Quantitative analysis of lattice disorder and crystallite size in organic semiconductor thin films. *Phys Rev B Condens Matter Mater Phys* **84**, 1–20 (2011).
3. Jacobs, I. E. *et al.* Structural and Dynamic Disorder, Not Ionic Trapping, Controls Charge Transport in Highly Doped Conducting Polymers. *J Am Chem Soc* **144**, 3005–3019 (2022).

4. Tjhe, D. H. L. *et al.* Non-equilibrium transport in polymer mixed ionic–electronic conductors at ultrahigh charge densities. *Nat Mater* **23**, 1712–1719 (2024).
5. Lee, K. *et al.* Metallic transport in polyaniline. *Nature* **441**, 65–68 (2006).
6. Kaiser, A. B. Systematic conductivity behavior in conducting polymers: Effects of heterogeneous disorder. *Advanced Materials* **13**, 927–941 (2001).
7. Sun, W., Du, A., Gao, G., Shen, J. & Wu, G. Graphene-templated carbon aerogels combining with ultra-high electrical conductivity and ultra-low thermal conductivity. *Microporous and Mesoporous Materials* **253**, 71–79 (2017).
8. Herrera-Ramírez, L. C., Cano, M. & Guzman de Villoria, R. Low thermal and high electrical conductivity in hollow glass microspheres covered with carbon nanofiber–polymer composites. *Compos Sci Technol* **151**, 211–218 (2017).
9. Liao, X. *et al.* Extremely low thermal conductivity and high electrical conductivity of sustainable carbon/ceramic electrospun nonwoven materials. *Sci Adv* **9**, eade6066 (2023).
10. Go, D. B. *et al.* Thermionic Energy Conversion in the Twenty-first Century: Advances and Opportunities for Space and Terrestrial Applications. *Front Mech Eng* **3**, 13 (2017).
11. O’Dwyer, M. F., Humphrey, T. E., Lewis, R. A. & Zhang, C. Low thermal conductivity short-period superlattice thermionic devices. *J Phys D Appl Phys* **39**, 4153–4158 (2006).
12. Bubnova, O. *et al.* Optimization of the thermoelectric figure of merit in the conducting polymer poly(3,4-ethylenedioxythiophene). *Nat Mater* **10**, 429–433 (2011).
13. Linseis, V., Völklein, F., Reith, H., Nielsch, K. & Woias, P. Advanced platform for the in-plane ZT measurement of thin films. *Review of Scientific Instruments* **89**, 015110 (2018).
14. Masoumi, S., Zhussupbekov, K., Prochukhan, N., Morris, M. A. & Pakdel, A. A comprehensive investigation into thermoelectric properties of PEDOT:PSS/Bi_{0.5}Sb_{1.5}Te₃ composites. *J Mater Chem C Mater* **12**, 14314–14329 (2024).
15. Butson, M. J., N Yu, P. K., Cheung -, T., Blanchard, M. & Feuilloy, M. Thermoelectric Performance of Poly(3,4-ethylenedioxythiophene): Poly(styrenesulfonate). *Chinese Phys. Lett* **25**, 2202 (2008).
16. Tsuchikawa, R. *et al.* Unique Thermoelectric Properties Induced by Intrinsic Nanostructuring in a Polycrystalline Thin-Film Two-Dimensional Metal–Organic

- Framework, Copper Benzenehexathiol. *Physica Status Solidi (A) Applications and Materials Science* **217**, 2000437 (2020).
17. Huang, X. *et al.* Superconductivity in a Copper(II)-Based Coordination Polymer with Perfect Kagome Structure. *Angewandte Chemie* **130**, 152–156 (2018).
 18. Huang, X. *et al.* A two-dimensional π -d conjugated coordination polymer with extremely high electrical conductivity and ambipolar transport behaviour. *Nat Commun* **6**, 7408 (2015).
 19. Toyoda, R. *et al.* Heterometallic Benzenehexathiolato Coordination Nanosheets: Periodic Structure Improves Crystallinity and Electrical Conductivity. *Advanced Materials* **34**, 2106204 (2022).
 20. Pan, Z. *et al.* Synthesis and structure of a non-van-der-Waals two-dimensional coordination polymer with superconductivity. *Nat Commun* **15**, 9342 (2024).
 21. Simoncelli, M., Marzari, N. & Mauri, F. Wigner Formulation of Thermal Transport in Solids. *Phys Rev X* **12**, 041011 (2022).
 22. Legenstein, L., Reicht, L., Wieser, S., Simoncelli, M. & Zojer, E. Heat transport in crystalline organic semiconductors: coexistence of phonon propagation and tunneling. *NPJ Comput Mater* **11**, 29 (2025).